# *bicoid* mRNA localises to the *Drosophila* oocyte anterior by random Dynein-mediated transport and anchoring

Vítor Trovisco[1,2], Katsiaryna Belaya[1,2†], Dmitry Nashchekin[1,2], Uwe Irion[1,2‡], George Sirinakis[1,2], Richard Butler[1], Jack J Lee[3], Elizabeth R Gavis[3], Daniel St Johnston[1,2*]

[1]The Gurdon Institute, University of Cambridge, Cambridge, United Kingdom; [2]Department of Genetics, University of Cambridge, Cambridge, United Kingdom; [3]Department of Molecular Biology, Princeton University, Princeton, United States

*For correspondence:
d.stjohnston@gurdon.cam.ac.uk

Present address: [†]The Weatherall Institute of Molecular Medicine, University of Oxford, Oxford, United Kingdom; [‡]Max Planck Institute for Developmental Biology, Tübingen, Germany

Competing interests: The authors declare that no competing interests exist.

**Abstract** *bicoid* mRNA localises to the *Drosophila* oocyte anterior from stage 9 of oogenesis onwards to provide a local source for Bicoid protein for embryonic patterning. Live imaging at stage 9 reveals that *bicoid* mRNA particles undergo rapid Dynein-dependent movements near the oocyte anterior, but with no directional bias. Furthermore, *bicoid* mRNA localises normally in *shot*[2A2], which abolishes the polarised microtubule organisation. FRAP and photo-conversion experiments demonstrate that the RNA is stably anchored at the anterior, independently of microtubules. Thus, *bicoid* mRNA is localised by random active transport and anterior anchoring. Super-resolution imaging reveals that *bicoid* mRNA forms 110–120 nm particles with variable RNA content, but constant size. These particles appear to be well-defined structures that package the RNA for transport and anchoring.

## Introduction

mRNA localisation is a widely-used mechanism for targeting proteins to the regions of the cell where they are required and is often coupled to translational repression to prevent expression of the encoded protein until after its transcript is localised (*Lécuyer et al., 2007*; *Jambor et al., 2015*). This is particularly important during axis formation in organisms such as *Drosophila* and *Xenopus* where mRNAs localise during oogenesis to provide the primary patterning signals for the embryo. In *Drosophila,* the anterior-posterior axis is determined by the microtubule-dependent localisation of *bicoid* (*bcd*) and *oskar* (*osk*) mRNAs to the anterior and posterior poles of the oocyte, respectively (*Pokrywka and Stephenson, 1991*; *Clark et al., 1994*; *Roth et al., 1995*). *bcd* mRNA is translationally repressed during oogenesis and is only translated when the egg is laid, providing a local source of Bcd protein, which diffuses to form a morphogen gradient that patterns the anterior half of the embryo (*Ephrussi and St Johnston, 2004*). By contrast, *osk* mRNA is translated when it reaches the posterior of the oocyte to produce long and short isoforms of Oskar protein (*Markussen et al., 1995*; *Rongo et al., 1995*). Long Oskar anchors its own RNA, whereas short Oskar nucleates the polar granules, leading to the posterior recruitment of the germ line determinants and the abdominal determinant, *nanos* mRNA (*Wang and Lehmann, 1991*; *Ephrussi and Lehmann, 1992*; *Vanzo and Ephrussi, 2002*).

Both *bcd* and *osk* mRNAs are transcribed in the nurse cells within the germline cyst and are then transported along microtubules through the ring canals into the oocyte by Dynein (*Clark et al., 2007*; *Mische et al., 2007*). The localisation of *osk* mRNA to the posterior of the oocyte requires the plus end-directed microtubules motor protein, Kinesin-I (*Brendza et al., 2000*). Live imaging of

**eLife digest** Molecules of messenger RNA, or mRNA for short, contain the instructions needed to make proteins. Many mRNAs are only found in certain parts of the cell to ensure that the corresponding proteins are only produced where they are actually needed. The mRNAs are delivered to their final location in the cell by motor proteins that move along tracks made of filaments called microtubules.

In female fruit flies, a mRNA called *bicoid* is transported to front end of a developing egg cell, while another mRNA called *oskar* is moved to the rear end. When the egg is fertilized, the region that contains *bicoid* mRNA develops into the head of the embryo, while the other end gives rise to the abdomen. A motor protein called kinesin-1 transports the *oskar* mRNA to the rear end, but how *bicoid* mRNA moves to the front end is not clear.

Trovisco et al. used microscopy to study how *bicoid* mRNA moves. The experiments show that another motor protein called Dynein moves *bicoid* mRNA along microtubules. However, unlike *oskar* mRNA, *bicoid* mRNA moves along microtubules in all directions and is not biased towards the front end of the cell. Trovisco et al. hypothesized that when *bicoid* mRNA reaches the front end of the egg it is trapped there by other factors.

Further experiments found that *bicoid* mRNA is indeed anchored at the front end of the cell. The mRNA does not seem to be trapped at the ends of the microtubules along which it is transported, nor does it form large clumps. Instead, it forms small, well-defined particles that remain the same size as the egg develops. The findings of Trovisco et al. raise the possibility that *bicoid* mRNA is packaged into these particles in order to be transported and anchored at the front end of the egg cell. Future work is needed to understand how particles containing *bicoid* mRNA are tethered at the front end of the egg cell and whether other mRNAs are also packaged in a similar manner.

fluorescently-labelled *osk* mRNA reveals that it forms particles that undergo rapid movements in all directions with a slight posterior bias, indicating that the RNA takes a biased random walk to the posterior cortex, where it is then anchored (*Zimyanin et al., 2008*). Since almost all *osk* mRNA movements depend on Kinesin-I, the microtubule cytoskeleton appears to be largely disordered, with a small excess of microtubule plus ends pointing posteriorly. This is consistent with measurements of the direction of growing microtubule plus ends, which reveal that most grow from the anterior/lateral cortex and extend in random directions with a weak orientation bias that is stronger close to the posterior pole (*Parton et al., 2011*).

How *bcd* mRNA is targeted to the anterior of the oocyte at mid-oogenesis is less well understood. Disrupting the Dynein/Dynactin complex by over-expressing Dynamitin causes either a posterior spreading or complete delocalisation of *bcd* mRNA, suggesting that the RNA is localised by Dynein-dependent, minus end-directed transport along microtubules (*Clark et al., 1997*; *Duncan and Warrior, 2002*; *Januschke et al., 2002*). However, as Dynein is also required for *bcd* mRNA transport into the oocyte, it is hard to distinguish direct from indirect effects. Furthermore, injected naïve *bcd* mRNA accumulates at the nearest region of the anterior/lateral cortex to its site of origin and not specifically at the anterior, consistent with the observation that microtubules ends are anchored or nucleated from all of the cortex except the very posterior pole (*Cha et al., 2001*). Only pre-treatment with nurse cell cytoplasm renders *in vitro* transcribed RNA competent to localise specifically to the oocyte anterior, and this conditioning requires the the pseudonuclease Exuperantia (Exu) (*Cha et al., 2001*). The role of Exu in the localisation of *bcd* mRNA requires its homo-dimerisation and RNA binding (*Lazzaretti et al., 2016*). Finally, computer simulations of the microtubule network in the anterior region of the oocyte suggest that it has little orientation bias, making it unlikely that RNA movement towards microtubules minus ends can account for the rapid anterior accumulation of the RNA (*Trong et al., 2015*).

The number of genes required for *bcd* mRNA localisation increases as oogenesis proceeds, suggesting distinct mechanisms localise the RNA at different stages. *exu* is required at all stages of localisation, whereas *swallow*, the γ-tubulin ring complex (γ-TURC), *staufen* and the ESCRT-II complex are only needed from stage 10b onwards (*Berleth et al., 1988*; *St Johnston et al., 1989*;

*Ferrandon et al., 1994*; *Irion and St Johnston, 2007*; *Schnorrer et al., 2000*, *2002*; *Weil et al., 2006*). The γ-TURC forms a new microtubule organising centre (MTOC) in the middle of the anterior cortex at this stage and this coincides with the re-localisation of *bcd* mRNA from an anterior ring into a central disc adjacent to the MTOC. Furthermore, live imaging of stage 10b-12 oocytes reveals that *bcd* mRNA particles move towards the anterior in a Dynein-dependent manner (*Weil et al., 2006*, *2008*). Since the RNA localisation is labile, it has been proposed that it is not anchored at the anterior at this stage and continually diffuses away, to be re-localised by Dynein-mediated transport (*Weil et al., 2008*).

Here we use fast live imaging, fluorescence recovery after photo-bleaching, photo-conversion and super-resolution microscopy to investigate the mechanism of the initial localisation of *bcd* mRNA to the anterior at stage 9. Our results indicate that at this stage the mRNA is not localised by continual directed transport, but by random active transport and anterior anchoring.

## Results

In order to visualise *bcd* mRNA in living oocytes, we took advantage of the MS2-system for fluorescently labelling RNA *in vivo*, in which the RNA contains multiple MS2 stem-loops that are bound by the MS2 coat protein (MCP) coupled to a fluorescent reporter (*Bertrand et al., 1998*; *Forrest and Gavis, 2003*; *Weil et al., 2008*). The original genomic *bicoid*-MS2 transgenes expressed full-length *bcd* mRNA from its endogenous promoter fused to 6 copies of the MS2 stem loop (*Weil et al., 2006*), but the relatively low numbers of MCP-GFP bound per RNA and the low expression levels made the RNA hard to image, particularly in fast moving particles. We therefore generated a construct in which the maternal α4 tubulin promoter drives the expression of the *bcd* 3'UTR fused to 11 copies of the MS2 stem-loop (*bcdMS2*), as the 3'UTR is sufficient for all steps in *bcd* RNA localisation (*Macdonald and Struhl, 1988*). The removal of the coding region allowed us to express this RNA at higher levels without disrupting embryonic development by expanding the Bcd morphogen gradient (*Namba et al., 1997*). Co-expression of *bcdMS2* with MCP-GFP (*bcd*\*GFP) gave strong labelling of *bcd* mRNA, which showed an identical localisation to the endogenous transcript at all stages of oogenesis (*Figure 1A–B*, data not shown).

### Kinesin-I antagonises the Dynein-dependent transport of *bcd* mRNA

Fast, high magnification wide-field imaging of *bcd*\*GFP in stage 9 oocytes revealed many small RNA particles that moved at speeds of up to 2.2 μm/sec (*Figure 1C–D*, *Video 1*). All movements were abolished by treatment with the microtubule-depolymerising drug, Colcemid, whereas the actin depolymeriser, Cytocholasin D, caused premature cytoplasmic streaming but had no effect on particle motility (*Figure 1—figure supplement 1A*, *Video 2*, data not shown). These results are consistent with the observation that *bcd* mRNA localisation at all stages of oogenesis is disrupted by microtubule-depolymerising drugs, and supports the view that particle movements play a role in delivering the mRNA to the oocyte anterior (*Pokrywka and Stephenson, 1991*; *Weil et al., 2006*).

The *bcd* mRNA particles in oocytes with one copy of *bcdMS2* moved with an average velocity of 0.64 μm/sec, which is significantly faster than *osk* mRNA particles (0.47 μm/sec) imaged under equivalent conditions (*Zimyanin et al., 2008*) (*Table 1*, *Table 1—source data 1*, *Figure 1D*). This difference was even more marked when we imaged egg chambers expressing two copies of *bcdMS2*, with the mean velocity increasing to 0.78 μm/s (*Table 1*, *Table 1—source data 1*, *Figure 1D*). This increase is presumably because the signal from the fastest particles is spread across more camera pixels per frame (6 pixels for particles moving at 2 μm/s, imaged for 0.25 s), making them harder to detect. Doubling their brightness therefore increases the efficiency of detection of the fastest particles.

The observation that *bcd* mRNA particles move at nearly twice the average speed of *osk* mRNA particles suggests that they are transported by different motor proteins. The most likely candidate for a motor that transports *bcd* mRNA is cytoplasmic Dynein, since putative microtubule minus end markers are enriched anteriorly (*Clark et al., 2007*, *1997*). It is not possible to test null mutations in Dynein components, as these block oocyte determination. We therefore used a combination of hypomorphic alleles of the Dynein heavy chain, $Dhc^{6-10}/Dhc^{8-1}$, that has previously been shown to reduce the speed of mRNA movement towards the microtubule minus ends in the embryo (*Bullock et al., 2006*). *bcd* mRNA particles moved with a mean velocity of 0.50 μm/sec in $Dhc^{6-10}/$

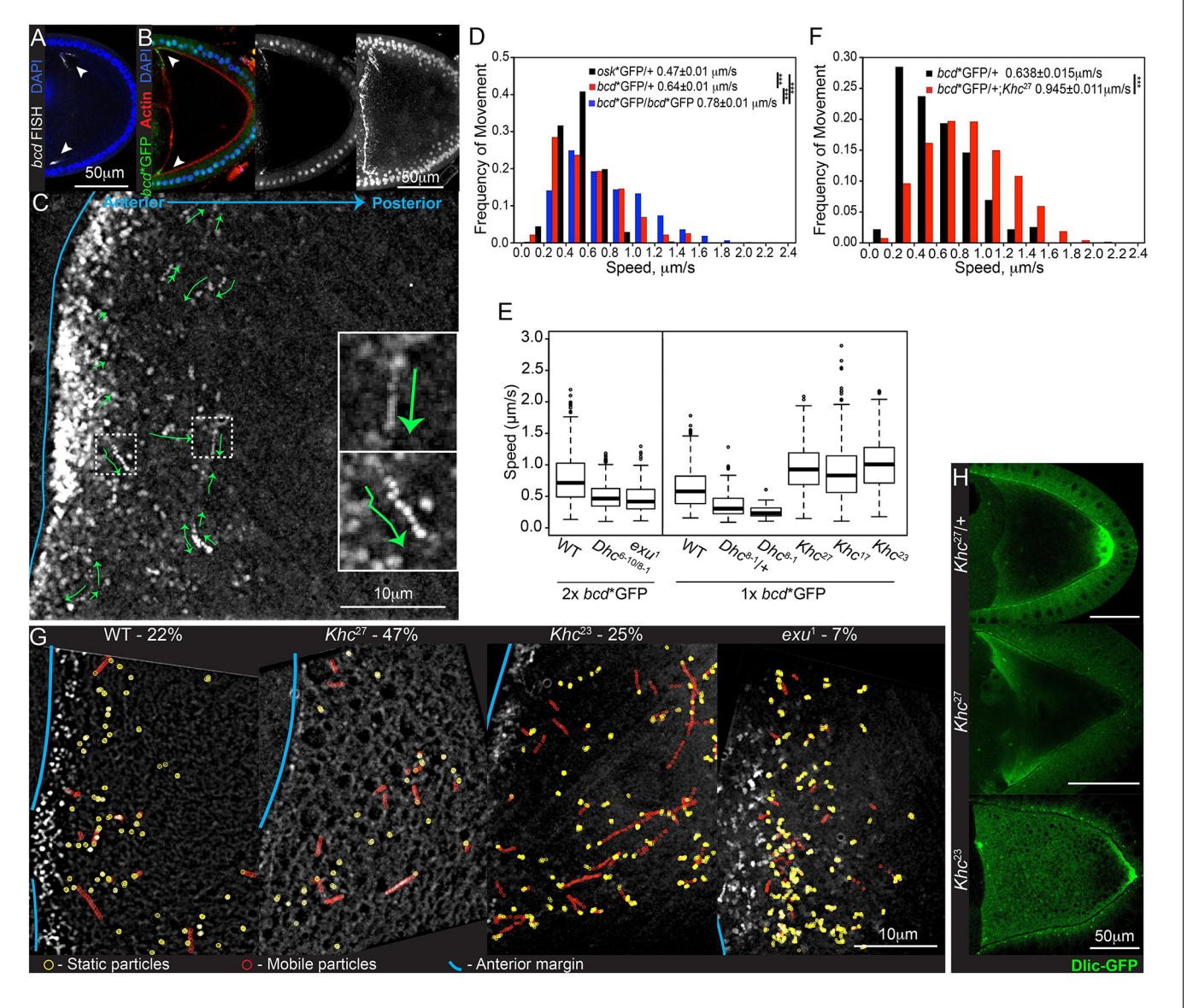

**Figure 1.** *bcd* mRNA is actively transported by cytoplasmic Dynein along microtubules. (**A–B**) Confocal microscopy images of stage 9 egg chambers showing that the endogenous (**A**; FISH) and transgenic *bcd* mRNA (**B**; *bcd*\*GFP) have the same anterior-lateral localization (arrowheads); DNA (DAPI) in blue. (**B**) Left – mid-sagittal plane, with *bcd*\*GFP in green and actin (Phalloidin-TRITC) in red; middle – mid-sagittal plane, with *bcd*\*GFP alone; right – maximum intensity projection over half of the oocyte volume, showing the anterior-lateral ring of *bcd*\*GFP. (**C**) Time-projection of high-magnification, wide-field live imaging of *bcd*\*GFP particles; the green arrows highlight moving particles; the blue line marks the oocyte anterior; insets are magnifications of the dashed white boxes. (**D**) Speed distribution of fast *osk* mRNA (black; *Zimyanin et al, 2008*) or *bcd* mRNA particles (red - one copy of *bcd*\*GFP; blue - two copies of *bcd*\*GFP). (**E**) Boxplot of the speeds of *bcd* mRNA particles in wild-type and mutant oocytes. (**F**) Speed distribution of fast *bcd* mRNA particles in wild-type and *Khc27* mutant oocytes. (**G**) Time-projection of high-magnification, wide-field live imaging of *bcd*\*GFP particles in wild-type and mutant oocytes; yellow circles highlight static particles, red circles highlight moving particles and the blue line indicates the oocyte anterior; the percentages show the mobile fraction of *bcd* mRNA particles during 5 s intervals. (**H**) Dlic-GFP in wild-type, *Khc27* and *Khc23* mutant stage 9 oocytes.

The following figure supplement is available for figure 1:

**Figure supplement 1.** bcdmRNA is actively transported by cytoplasmic Dynein along microtubules.

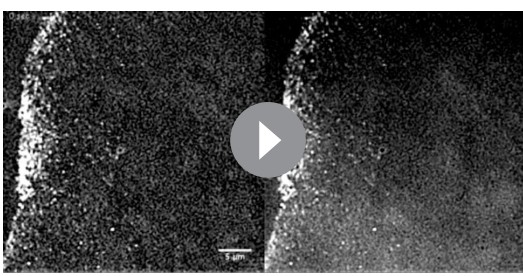

**Video 1.** (related to *Figure 1C*) – *bcd* mRNA assembles into particles that undergo fast active transport. High-magnification, wide-field live imaging of *bcd*\*GFP in stage 9 oocytes. The right panel shows the fast moving RNA particles as coloured tracks. Images were acquired at a rate of 0.64 s/frame.

$Dhc^{8-1}$ compared with 0.78 µm/sec in wild-type oocytes (*Table 1*, *Table 1—source data 1*, *Figure 1E*, *Figure 1—figure supplement 1C*). We observed a similar reduction in the velocities of *bcd* mRNA particles in $Dhc^{8-1}$/+ heterozygotes (59% of wild-type; *Table 1*, *Table 1—source data 1*, *Figure 1E*, *Figure 1—figure supplement 1C*), suggesting that $Dhc^{6-10}$ has little effect on motor speed and that $Dhc^{8-1}$ has a dominant negative effect. Consistent with this, homozygous mutant germline clones of $Dhc^{8-1}$ showed an even greater reduction in particle velocity to 39% of the wild-type speed (*Table 1*, *Table 1—source data 1*, *Figure 1E*, *Figure 1—figure supplement 1C*). Thus, this allele produces a functional motor protein that still moves, but significantly more slowly than the wild-type protein. The strong reduction in the speed of *bcd* mRNA particles in $Dhc^{8-1}$ homozygotes indicates that the majority are transported by Dynein. The $Dhc^{8-1}$ mutant also significantly reduces the amount of *bcd* mRNA localised to the anterior (*Figure 1—figure supplement 1D*). This does not seem to be due to reduced frequency of movement because the mobile fraction of *bcd* mRNA particles is unaffected in $Dhc^{6-10}$/$Dhc^{8-1}$ mutants (26% in wild-type *versus* 22% in mutant; *Table 2*, *Table 2—source data 1*, *Figure 1G*). Thus, slower Dynein-dependent transport, presumably both from the nurse cells into the oocyte and within the oocyte, impairs the delivery of *bcd* mRNA to the oocyte anterior.

Null mutations in the Kinesin heavy chain (*Khc*) also disrupt the localisation of *bcd* mRNA, with the majority of oocytes showing spreading of the RNA along the anterior and lateral cortex (*Januschke et al., 2002*) (*Figure 1—figure supplement 1E*). It is unclear whether this phenotype arises because Kinesin-I plays a direct role in the transport and/or anchoring of *bcd* mRNA. Kinesin-I transports Dynein to the oocyte posterior, indicating that the two motors can associate in the same complex, and Kinesin-I could therefore affect *bcd* mRNA indirectly, for example by recycling Dynein to the oocyte posterior for further rounds of minus end-directed transport, or by modulating the activity of Dynein (*Januschke et al., 2002*; *Palacios and St Johnston, 2002*). To test the role of Kinesin-I directly, we analysed the movement of *bcd* mRNA particles in germline clones of the null allele, $Khc^{27}$(*Brendza et al., 2000*). Surprisingly, the velocity of *bcd* mRNA particle movements was significantly increased in the absence of Kinesin-I, with an average speed of 0.98 µm/sec, compared to 0.64 µm/sec in wild-type (*Table 1*, *Table 1—source data 1*, *Figure 1E–F*, *Figure 1—figure supplement 1C*). This increase could be explained if a fraction of the *bcd* mRNA particles are transported at low speeds by Kinesin-I, so that its loss raises the average velocity by removing the slow population of moving particles. If so, one would expect the fraction of mobile particles to be reduced in $Khc^{27}$. In wild-type, approximately 20% of the *bcd* mRNA particles move over a five second period (*Table 2*, *Table 2—source data 1*, *Figure 1G*), which is nearly twice the proportion observed for *osk* mRNA (*Zimyanin et al., 2008*). In $Khc^{27}$ mutants, the mobile fraction more than doubled to 47% (*Table 2*, *Table 2—source data 1*, *Figure 1G*). Thus, both the speed and frequency of *bcd* mRNA particle motility are increased in the absence of Kinesin-I, making it highly unlikely that this motor is responsible for a significant

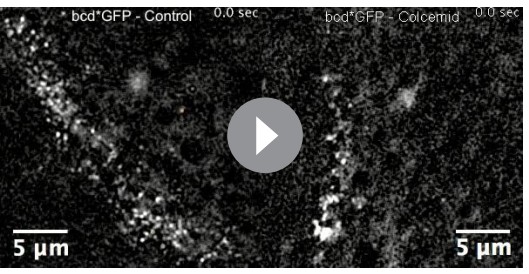

**Video 2.** (related to *Figure 1—figure supplement 1C*) – *bcd* mRNA particles undergo microtubule-dependent active transport. High-magnification, wide-field live imaging of *bcd*\*GFP in stage 9 oocytes, with and without depolymerisation of microtubules. Left – mock; Right – Colcemid (400 µg/ml). The fast moving RNA particles are shown as coloured tracks. Images were acquired at a rate of 0.18 s/frame.

**Table 1.** Parameters of fast *bcd* mRNA in wild-type and mutant oocytes.

| Genotype (2x *bcd*\*GFP) | Tracks (n) | Movies (n) | Oocytes (n) | Anterior Mov (%/ n) | Binomial P-value [a)] | Speed ±S.E.M. (µm/s) | Wilcoxon P-value [b)] | Mixed-effects P-value [c)] | Track distance ±S.E.M. (µm) | Anterior displacement ±S.E.M. (µm/s) | Wilcoxon P-value [d)] |
|---|---|---|---|---|---|---|---|---|---|---|---|
| Wild-type | 1181 | 32 | 9 | 52.6 / 621 | 0.040 * | 0.78 / 0.01 | | | 1.41 / 0.03 | 0.02 / 0.02 | 0.200 |
| *Dhc*[6-10/8-1] | 669 | 7 | 5 | 50.8 / 340 | 0.350 | 0.50 / 0.01 | <2.2E-16 *** | 6.90E-10 *** | 1.61 / 0.05 | 0.01 / 0.01 | 0.500 |
| *exu*[1] | 215 | 17 | 4 | 53.0 / 114 | 0.207 | 0.47 / 0.02 | <2.2E-16 *** | 3.10E-09 *** | 1.39 / 0.07 | 0.03 / 0.02 | 0.230 |

| Genotype (1x *bcd*\*GFP) | Tracks (n) | Movies (n) | Oocytes (n) | Anterior Mov (%/ n) | Binomial P-value [a)] | Speed ±S.E.M. (µm/s) | Wilcoxon P-value [b)] | Mixed-effects P-value [c)] | Track distance ±S.E.M. (µm) | Anterior displacement ±S.E.M. (µm/s) | Wilcoxon P-value [d)] |
|---|---|---|---|---|---|---|---|---|---|---|---|
| Wild-type | 450 | 22 | 13 | 54.4 / 245 | 0.033 * | 0.64 / 0.01 | | | 0.98 / 0.05 | 0.02 / 0.02 | 0.135 |
| *Khc*[27] GLC | 921 | 14 | 6 | 53.7 / 495 | 0.013 * | 0.94 / 0.01 | <2.2E-16 *** | 1.60E-06 *** | 1.55 / 0.04 | 0.01 / 0.02 | 0.272 |
| *Khc*[17] GLC | 639 | 21 | 12 | 53.2 / 340 | 0.057 | 0.89 / 0.02 | <2.2E-16 *** | 2.00E-04 *** | 1.60 / 0.05 | 0.02 / 0.02 | 0.401 |
| *Khc*[23] GLC | 612 | 15 | 5 | 53.9 / 330 | 0.029 * | 1.01 / 0.02 | <2.2E-16 *** | 2.20E-06 *** | 1.67 / 0.06 | 0.06 / 0.03 | 0.044 * |
| *Dhc*[8-1]/+ | 141 | 7 | 4 | - - - | - - - | 0.38 / 0.02 | 3.61E-16 *** | 3.00E-04 *** | 1.15 / 0.08 | 0.06 / 0.02 | - - - |
| *Dhc*[8-1] GLC | 43 | 16 | 7 | - - - | - - - | 0.25 / 0.02 | <2.2E-16 *** | 3.10E-07 *** | 0.72 / 0.06 | -0.05 / 0.03 | - - - |

a) Binomial test for the frequency of anterior-directed movements being >50% (one-tailed)

b) Wilcoxon rank sum test for speed comparisons - comparison to wild-type (2x *bcd*\*GFP or 1x *bcd*\*GFP)

c) Mixed-effects linear model (LMER) test for speed comparisons – comparison to wild-type (2x *bcd*\*GFP or 1x *bcd*\*GFP). Fixed Effect: Genotype; Random Effects: Variability between oocytes and movies

d) Wilcoxon 1-sample test for the net anterior displacement. Null hypothesis: mean=0 (two-tailed)

*p<0.05; **p<0.01; ***p<0.001

- - - Not applicable / Not done

Source data 1. Tracking of *bcd*\*GFP particles in wild-type and mutant stage 9 oocytes. Includes the data in: **Table 1**; **Figure 1** panels D–F; **Figure 2**, panels A–E; **Figure 2—figure supplement 1**, panel C.

proportion of the movements. Furthermore, *bcd* mRNA particles showed virtually no reversals of movement in either wild-type or *Khc*[27] mutants (1.9% *versus* 1.5%, respectively), suggesting the RNA does not alternate between transport by motors of opposing polarity.

Another way to test the role of Kinesin-I is to analyse two mutants, *Khc*[17] and *Khc*[23], with single amino acid changes in the motor domain that reduce the speed of Kinesin-I movement without affecting its other properties (**Brendza et al., 1999**; **Serbus et al., 2005**; **Zimyanin et al., 2008**). Like the null allele, germline clones of *Khc*[17] and *Khc*[23] also caused a significant increase in the mean velocity of *bcd* mRNA particle movements (0.89 µm/sec and 1.0 µm/sec, respectively) (**Table 1**, **Table 1—source data 1**, **Figure 1E**). This rules out the possibility that Kinesin-I is responsible for the slow movements of *bcd* mRNA particles, as this would result in a decrease in the average velocity in the mutants. Kinesin-I must therefore act by some other mechanism to reduce the speed of movement by another motor, presumably Dynein, for example by engaging in a tug of war. The slow allele, *Khc*[23], has little effect on the fraction of mobile particles (25%), compared to 47% in the null allele (**Table 2**, **Table 2—source data 1**, **Figure 1G**), indicating that the effects on speed and frequency are separable. One possible explanation for this difference is that the slow allele still transports the Dynein/Dynactin complex to the posterior pole of the oocyte, albeit more slowly, whereas the null allele does not (**Januschke et al., 2002**) (**Figure 1H**). The absence of Dynein transport to the

**Table 2.** Mobile fraction of *bcd* mRNA particles in wild-type and mutant oocytes.

| Genotype | Oocytes | Mobile fraction/5 s | T-test P-value | a) |
|---|---|---|---|---|
| *bcd*\*GFP/+ | 5 | 0.22 ± 0.03 | | |
| *bcd*\*GFP/+;*Khc*$^{27}$ GLC | 5 | 0.47 ± 0.02 | 0.0004 | *** |
| *bcd*\*GFP/+;*Khc*$^{23}$ GLC | 4 | 0.25 ± 0.02 | 0.624 | |
| *bcd*\*GFP/*bcd*\*GFP;*Dhc*$^{6-10/8-1}$ | 4 | 0.26 ± 0.05 | 0.556 | |
| *bcd*\*GFP/*bcd*\*GFP;*exu*$^{1}$ | 4 | 0.07 ± 0.01 | 0.006 | ** |

a) T-test for comparison of mobile fractions (two-tailed) - comparisons to *bcd*\*GFP/+ (wild-type)
**p<0.01; ***p<0.001
Source data 1. Mobile fraction of *bcd*\*GFP particles in wild-type and mutant stage 9 oocytes. Includes the data in:
*Table 2*; *Figure 1*, panel G.

posterior in the null mutant will therefore increase the concentration of Dynein at the anterior of the oocyte, which could account for the more than doubling of the frequency of particle movement in this region.

The localisation of *bcd* mRNA at all stages depends on Exu protein (*Berleth et al., 1988*) and we therefore also examined the behaviour of *bcd* mRNA particles in the mutant, *exu*$^{1}$, which reduces the affinity of Exu to RNA (*Lazzaretti et al., 2016*). Very few particles were visible in *exu*$^{1}$ homozygous oocytes, and these moved significantly less frequently than normal (7%) and at a reduced speed (*Table 1*, *Table 1—source data 1*, *Table 2*, *Table 2—source data 1*, *Figure 1E,G*, *Figure 1—figure supplement 1C*). Thus, Exu is required for both the formation of *bcd* mRNA particles and their efficient transport on microtubules, consistent with the dimerisation of Exu (possibly leading to the dimerisation of *bcd* mRNA) and the results obtained from *bcd* mRNA injections (*Cha et al., 2001*; *Lazzaretti et al., 2016*). The residual *bcd* mRNA motility in the mutant may explain why a small amount of RNA is still diffusely localised at the anterior of *exu* mutant oocytes (*Figure 1—figure supplement 1F*).

## Lack of a strong directional bias in *bcd* mRNA particle movement

We next assessed the directionality of *bcd* mRNA particle movements to determine if it could account for the anterior accumulation of the mRNA. We found only a slight excess of movements towards the anterior compared to the posterior over a region up to 40 μm from the anterior of the oocyte (52.6% *versus* 47.4%, p=0.04) (*Figure 2A*). The particles were tracked on near-surface optical sections, where they were better detected, but a similarly weak directional bias was also observed in deeper optical sections (*Figure 2—figure supplement 1A–B*). The velocity of the movements in each direction was not significantly different, and the net displacement was also not significantly different from zero (*Table 1*, *Table 1—source data 1*, *Figure 2A–C*). To test whether this bidirectional movement reflected motors moving in opposite directions along a strongly polarised microtubule cytoskeleton, or mainly unidirectional transport along a weakly polarised cytoskeleton, we measured the velocity in each direction in slow Dynein mutants. All *Dhc* mutant combinations reduced the velocity of posterior movements to the same extent as of anterior movements (*Figure 2B,D*). Thus, Dynein is responsible for the majority of particle movements both towards and away from the anterior cortex, and the absence of a strong bias in the direction of *bcd* mRNA transport is due to the very weak polarisation of the microtubule cytoskeleton in this anterior region. This is in good agreement with tracking of plus-ends of microtubules (*Parton et al., 2011*) and computer simulations of the oocyte microtubule network, which predict almost no bias in the orientation of microtubules near the anterior and a stronger bias in the posterior (*Trong et al., 2015*). The *exu* mutant caused a similar reduction in speed in both directions (*Figure 2B*), whereas *Khc* mutants increased the speed in both directions, consistent with the unpolarised nature of the microtubule network at the anterior (*Figure 2D–E*).

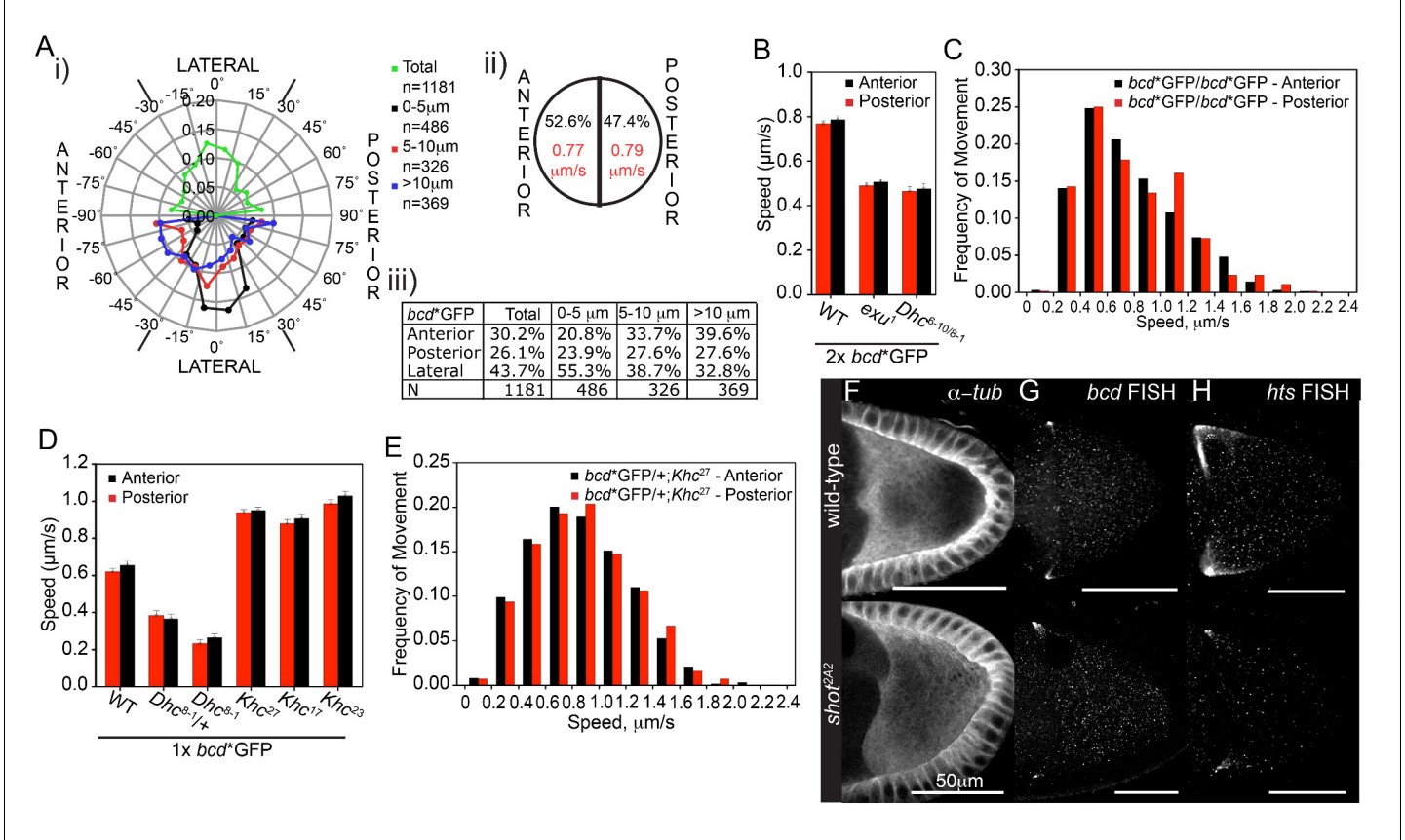

**Figure 2.** Fast *bcd* mRNA particles have little directional bias towards the oocyte anterior. (**A**) Directionality of the fast *bcd*\*GFP particles imaged near the cortex of stage 9 oocytes; i) Windchart of the frequency of movements per angle interval; the upper semi-circle shows all particles whereas the lower semi-circle shows particles according to their distance from the oocyte anterior; ii) Frequency and average speed of *bcd*\*GFP particles moving towards the anterior or posterior of the oocyte; iii) Frequency table of *bcd*\*GFP particles moving in anterior, posterior or lateral directions. (**B–C**) Average speed (mean ± S.E.M., 9 oocytes) (**B**) and speed distribution (**C**) of *bcd*\*GFP particles moving towards the anterior (black bar) or posterior (red bar) of wild-type oocytes. (**D–E**) Average speed (mean ± S.E.M., 6 oocytes) (**D**) and speed distribution (**E**) of *bcd*\*GFP particles moving towards the anterior (black bar) or posterior (red bar) of *Khc*[27] mutant oocytes. (**F–H**) Confocal images of microtubules (α-tub; **F**), endogenous *bcd* mRNA (RNA FISH; **G**) or endogenous *hts* mRNA (RNA FISH; **H**) in wild-type and *shot*[2A2] mutant oocytes.

The following source data and figure supplement are available for figure 2:

**Source data 1.** Tracking of *bcd*\*Tom particles in wild- stage 9 oocytes.

**Figure supplement 1.** Fast *bcd* mRNA particles have little directional bias towards the oocyte anterior.

We classified the particle movements according to their distance from the anterior cortex to determine if the bias varies across this region (*Figure 2A*). This revealed that there is a strong anterior bias of 12% in the movements of particles that are more than 10 μm from the anterior cortex (39.6% *versus* 27.6%), but this decreases to 6% in the region 5–10 μm from the anterior and reverses in the 0-5 μm region (20.8% *versus* 23.9%) to give a 3% excess of movements away from the anterior (*Figure 2A*). Similar results were obtained when tracking *bcd* mRNA particles deeper in the oocyte (*Figure 2—figure supplement 1A–B*, *Figure 2—source data 1*). Our analysis therefore indicates that although Dynein-dependent transport is required for *bcd* mRNA localisation, it is not sufficient to explain its robust anterior accumulation, because Dynein moves the RNA in and out of the anteriormost region at similar rates. This is incompatible with a model in which *bcd* mRNA is maintained by continuous anteriorly-directed transport, as has been proposed to occur at later stages of oogenesis (*Weil et al., 2006*). Instead, the data suggest that bidirectional transport facilitates the delivery

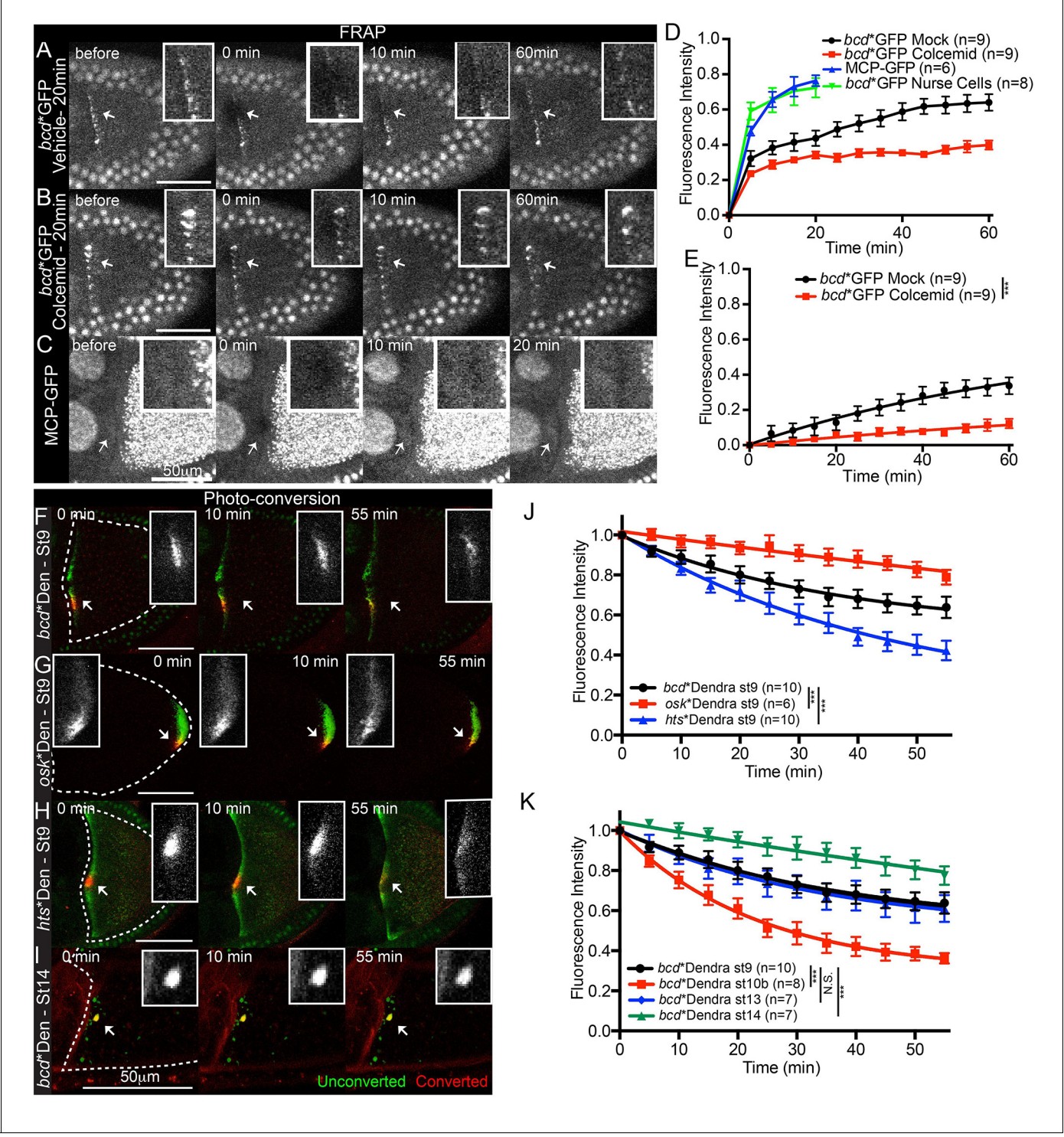

**Figure 3.** Localised *bcd* mRNA is anchored at the anterior of the oocyte. (A–E) Confocal time-series of FRAP experiments at the anterior of stage 9 oocytes (A–C) and the corresponding fluorescence recovery curves (D–E). (A–B) Egg chambers expressing *bcd*\*GFP were treated with Colcemid (B, 400 µg/ml) or control vehicle alone (A) 20 min prior to photobleaching. (C) Egg chamber expressing only MCP-GFP. (D–E) Graphs of FRAP of *bcd*\*GFP or MCP-GFP alone, before (D) or after (E) removal of the fast-recovering, nonspecific component. (F–K) Confocal time-series of photo-converted localised *bcd*\*Dendra2 (F,I), *osk*\*Dendra2 (G) and *hts*\*Dendra2 (H) and the corresponding fluorescence decay graphs after removal of the fast-recovering, nonspecific component (J–K). Dashed lines mark the outline of the oocyte; arrows indicate the photobleached or photo-converted regions and the insets are the corresponding close-ups. \*\*\* F-test P value <0.0001. N.S. Statistically not significant.

*Figure 3 continued on next page*

*Figure 3 continued*

The following source data and figure supplement are available for figure 3:

**Source data 1.** Photo-conversion data for *bcd*\*Dendra2 and *grk*\*Dendra2 (only timepoints 10 min and 60 min after photo-conversion).
**Figure supplement 1.** Localised *bcd* mRNA is anchored at the anterior of the oocyte.

of *bcd* mRNA particles to the anterior where they are specifically sequestered by some other mechanism.

To further test whether *bcd* mRNA localisation is independent of a polarised microtubule cytoskeleton, we examined the phenotype of *shot*[2A2] (*Chang et al., 2011*). *shot*[2A2] disrupts the anchoring of microtubules to the anterior/lateral cortex of the oocyte, resulting in a largely unpolarised microtubule network and the failure to localise *osk* mRNA to the posterior (*Figure 2F*) (*Nashchekin et al., 2016*). *bcd* mRNA localises normally in *shot*[2A2] mutant oocytes, despite the lack of directional bias in microtubule orientation (*Figure 2G*). For comparison, we analysed the behaviour of *hu-li tai shao* (*hts*) mRNA, which also localises anteriorly, but to a somewhat broader region than *bcd* mRNA and with different genetic requirements (*Ding et al., 1993*). Unlike *bcd* mRNA, the anterior enrichment of *hts* mRNA is largely lost in *shot*[2A2] (*Figure 2H*). Thus, *hts* mRNA localises by a different mechanism to *bcd* mRNA that depends on the weakly polarised microtubule cytoskeleton in the vicinity of the anterior.

## Anterior anchoring of *bcd* mRNA

The proposal that there is a mechanism that retains or anchors *bcd* mRNA once it reaches the anterior predicts that the mRNA should be relatively stable at the anterior, whereas the continual transport model predicts a rapid turn-over of the localised RNA. To distinguish between these possibilities, we performed Fluorescent Recovery After Photobleaching (FRAP) experiments on localised *bcd* mRNA in stage 9 oocytes. The rate of recovery was best fit by two exponential curves, suggesting the existence of fast and slow recovering populations (*Figure 3A,D*, *Video 3*). Furthermore, only the slow population was affected by microtubule-depolymerisation with Colcemid (*Figure 3B, D*, *Video 3*), which abolishes the active transport of *bcd* mRNA particles (*Figure 1—figure supplement 1A*, *Video 2*). The fast population is therefore likely to correspond to highly-diffusive, nonspecific signal, most likely from autofluorescent background and/or free MCP-GFP. Consistent with this, FRAP on the cytoplasm of nurse cells, which have very low levels of *bcd* mRNA, or at the anterior of oocytes expressing MCP-GFP alone, yielded very fast recoveries that fit single exponential curves (*Table 3*, *Table 3—source data 1*, *Figure 3C–D*). We therefore used the recovery in the nurse cells to fit the FRAP data to a bi-exponential and then removed the nonspecific, fast component (see Material and Methods). The remaining specific signal recovered to 33% over an hour in untreated oocytes, a value that is reduced to 10% by Colcemid treatment (*Table 3*, *Table 3—source data 1*,

**Table 3.** FRAP kinetics of localised *bcd* mRNA.

| Sample | Mobile fraction @ 20 min | Fluorescence Half-time (min) | Oocyte (n) | |
|---|---|---|---|---|
| MCP-GFP - St9 | 0.76 | 3.7 | 6 | |
| *bcd*\*GFP / Nurse cell - St9 | 0.71 | 2.0 | 8 | |
| Sample | Mobile fraction @ 55 min | Fluorescence Half-time (min) | Oocytes (n) | F-test P-value a) |
| *bcd*\*GFP - St9 - Mock | 0.33 | 52.2 | 9 | |
| *bcd*\*GFP - St9 - Colcemid | 0.10 | 75.6 | 9 | <0.0001 |

a) F-test for pairwise comparison of fluorescence recovery curves - comparisons to *bcd*\*GFP/+ (wild-type)
Source data 1. FRAP data for MCP-GFP and *bcd*\*GFP in stage 9 oocytes.
Includes the data in: *Table 3*; *Figure 3*, panels D–E.

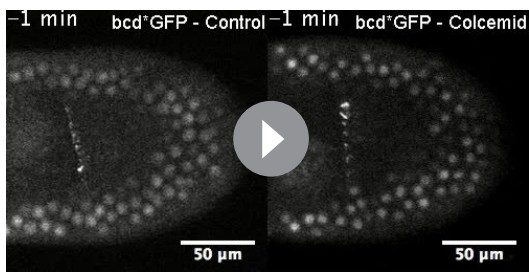

**Video 3.** (related to *Figure 3A–B*) – Anteriorly-localised *bcd* mRNA has slow and limited turn-over. FRAP of anteriorly-localised *bcd*\*GFP in stage 9 oocytes, with and without depolymerisation of microtubules. Left – Control; Right – Colcemid (400 µg/ml). Confocal images were acquired every 5 min, for 60 min.

*Figure 3E*). Thus, most *bcd* mRNA is stably anchored at the anterior cortex at stage 9, and the limited recovery is predominantly due to microtubule-dependent delivery of mRNA.

The FRAP experiments cannot distinguish whether the recovery is due to the *de novo* delivery of newly-synthesised *bcd* mRNA or to the recycling of previously localised RNA from outside the bleached region, as predicted by the continual transport model. To distinguish between these possibilities, we generated a transgene expressing MCP fused to the photo-convertible protein Dendra2 (MCP-Dendra2) so that we could label only the RNA that is already localised (*Gurskaya et al., 2006*; *Chudakov et al., 2007*). 63% of photo-converted *bcd* mRNA (*bcd*\*Dendra2) remained localised in the same small region over a 55 min period, in good agreement with the FRAP data (*Table 4*, *Table 4—source data 1*, *Figure 3F,J*, *Video 4*).

Moreover, there was very little spreading of the photo-converted RNA along the anterior margin, arguing against continual re-localisation of the RNA (*Figure 3F*).

We also analysed the behaviour of MS2-tagged *hts* mRNA labelled with MCP-Dendra2 (*hts*\*Dendra2). Unlike *bcd* mRNA, photo-converted *hts*\*Dendra2 showed marked spreading along the anterior cortex and was significantly more labile, with less than half (41%) of the signal remaining by the end of the time course (*Figure 3H,I*, *Video 5*). Both the spreading and the lower retention of localised *hts* mRNA are consistent with the idea that its anterior enrichment depends on continual transport, unlike *bcd* RNA at stage 9 of oogenesis.

As *bcd* RNA has been proposed to be localised by continual active transport beginning at stage 10b, we examined if the retention of localised *bcd* mRNA varies during oogenesis, by performing similar photo-conversion experiments on stage 10b, stage 13 and stage 14 oocytes. The RNA is significantly more mobile at stage 10b, with only 36% remaining in the ROI after 55 min (*Figure 3K*), consistent with the results of *Weil et al. (2006)*. Stage 13 showed similar levels of retention to stage 9, but the RNA was significantly more stable at stage 14, coincident with the formation of larger aggregates (*Figure 3J–K*) (*Weil et al., 2008*). *bcd* mRNA therefore appears to be localised by a distinct mechanism at stage 10b, when fast cytoplasmic streaming starts and the RNA relocalises from an anterior ring into a central disc (*Theurkauf et al., 1992*; *Schnorrer et al., 2000*, *2002*).

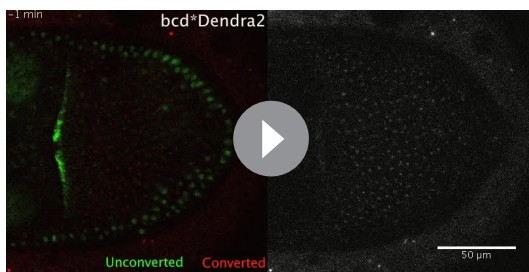

**Video 4.** (related to *Figure 3F*) – Localised *bcd* mRNA is stably anchored at the oocyte anterior. Photo-conversion of anteriorly-localised *bcd*\*Dendra2 in a stage 9 oocyte. Left – Unconverted *bcd*\*Dendra2 in green, photo-converted in red; Right – Photo-converted *bcd*\*Dendra2 alone. Confocal images were acquired every 5 min, for 55 min.

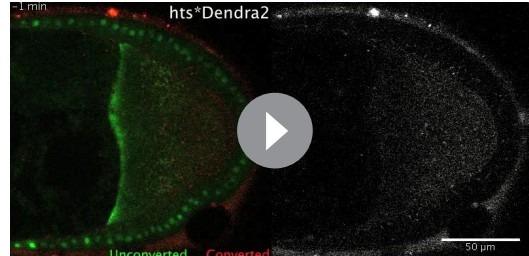

**Video 5.** (related to *Figure 3H*) – Localised *hts* mRNA is less stable at the oocyte anterior and spreads laterally. Photo-conversion of anteriorly-localised *hts*\*Dendra2 in a stage 9 oocyte. Left – Unconverted *hts*\*Dendra2 in green, photo-converted in red; Right – Photo-converted *hts*\*Dendra2 alone. Confocal images were acquired every 5 min, for 55 min.

**Table 4.** Photo-conversion kinetics of localised mRNAs.

| Sample | Mobile fraction @ 20 min | Fluorescence Half-time (min) | Oocytes (n) | | |
|---|---|---|---|---|---|
| *bcd*\*Dendra2 / Nurse cell - St9 | 0.93 | 3.2 | 10 | | |
| mRNA | Mobile fraction @ 55 min | Fluorescence Half-time (min) | Oocytes (n) | F-test P value | |
| *bcd*\*Dendra2 - St9 | 0.37 | 26.0 | 10 | | |
| *osk*\*Dendra2 - St9 | 0.20 | 173.6 | 5 | <0.0001 | a) |
| *hts*\*Dendra2 - St9 | 0.59 | 31.7 | 10 | <0.0001 | a) |
| *bcd*\*Dendra2 - St10 | 0.64 | 16.1 | 8 | <0.0001 | a) |
| *bcd*\*Dendra2 - St13 | 0.39 | 24.5 | 7 | 0.81 | a) |
| *bcd*\*Dendra2 - St14 | 0.24 | 138.5 | 10 | <0.0001 | a) |
| *bcd*\*Dendra2 / *grk*$^{2E12/2B6}$ - St9 | 0.40 | 33.2 | 11 | 0.91 | a) |
| *bcd*\*Dendra2 / 2x *bcdMS2* - St9 | 0.44 | 51.4 | 5 | 0.4 | a) |
| mRNA | Mobile fraction @ 55 min | Fluorescence Half-time (min) | Oocytes (n) | F-test P value | |
| *bcd*\*Dendra2 - St9 - Mock | 0.35 | 27.9 | 5 | 0.86 | a) |
| *bcd*\*Dendra2 - St9 - Colcemid | 0.22 | 152.1 | 5 | <0.0001 | a),b) |

a),b) F-test for pairwise comparison of fluorescence recovery curves. Comparisons to (a) *bcd*\*Dendra2 - St9 or (b) *bcd*\*Dendra2 - St9 - Mock

**Source data 1.** Photo-conversion data for all samples (MCP-Dendra2; *bcd*\*Dendra2, *osk*\*Dendra2 and *hts*\*Dendra2 in wild-type stage 9 oocytes; *bcd*\*Dendra2 in stages 10b, 13 and 14 wild-type oocytes; *bcd*\*Dendra2 in wild-type stage9 oocytes expressing 2 copies of the *bcdMS2* transgene; *bcd*\*Dendra2 in *grk* mutant stage9 oocytes; *bcd*\*Dendra2 in wild-type stage 9 oocytes treated with Colcemid or mock Control). Includes the data in: *Table 4*; *Figure 3*, panels J-K; *Figure 3—figure supplement 1*, panels B, F; *Figure 4*, panel C.

Since our results suggest that *bcd* mRNA is anchored in some way at the anterior of the oocyte at stage 9, we compared its behaviour to that of *osk* and *gurken* (*grk*) mRNAs, which are both anchored to the cytoskeleton after their localisation (*Delanoue et al., 2007*; *Vanzo and Ephrussi, 2002*). Although *osk* RNA is more stably anchored than *bcd* mRNA, with 80% of the signal remaining at the end of the experiment (*Figure 3G,I*), *grk* and *bcd* mRNAs showed almost identical retention rates (*Figure 3—figure supplement 1C–D*, *Figure 3—source data 1*). The greater retention of *osk* mRNA compared to *bcd* and *grk* mRNAs could reflect distinct anchoring mechanisms, but might also be due to different conditions at the posterior of the oocyte relative to anterior, where *bcd* and *grk* RNAs localise. To directly test of the effects of position within the oocyte, we examined *bcd* mRNA stability in strong *grk* mutants, in which *bcd* mRNA localises to both anterior and posterior poles (*González-Reyes et al., 1995*; *Roth et al., 1995*). Photo-converted *bcd* mRNA at the posterior of *grk* mutants yielded decay kinetics indistinguishable from those of RNA at the anterior of wild-type oocytes (*Figure 3—figure supplement 1E–F*, *Table 4*, *Table 4—source data 1*). Thus, the stability of localised *bcd* mRNA is intrinsic and not a consequence of the local geometry of the oocyte.

We next investigated mechanisms that might retain *bcd* mRNA at the anterior cortex. *grk* mRNA is anchored at the dorsal anterior corner of the oocyte by the binding of static Dynein to minus ends of microtubules, a mechanism that also anchors pair rule transcripts apically in the blastoderm embryo (*Delanoue and Davis, 2005*; *Delanoue et al., 2007*). We therefore tested whether the anterior retention of *bcd* mRNA is microtubule-dependent by culturing the egg chambers in the presence of Colcemid. Although the microtubules were completely depolymerised after ten minutes, as monitored with the microtubule binding protein, Tau-GFP, *bcd* mRNA labelled with MCP-Tomato (*bcd*\*Tom) remained tightly localised after 1 hr (*Figure 4A*, *Video 6*). Furthermore, performing the same experiment with photo-converted RNA revealed that the immobile fraction increased from 65% to 78% in the absence of microtubules (*Table 4*, *Table 4—source data 1*, *Figure 4B–C*). Thus, the anchoring of *bcd* mRNA is microtubule-independent, and microtubule-dependent processes enhance its depletion from the anterior cortex. It is notable that *bcd* mRNA is as stably localised as *osk* mRNA in the absence of microtubules, with only about 20% loss over the period of 55 min. *hts*

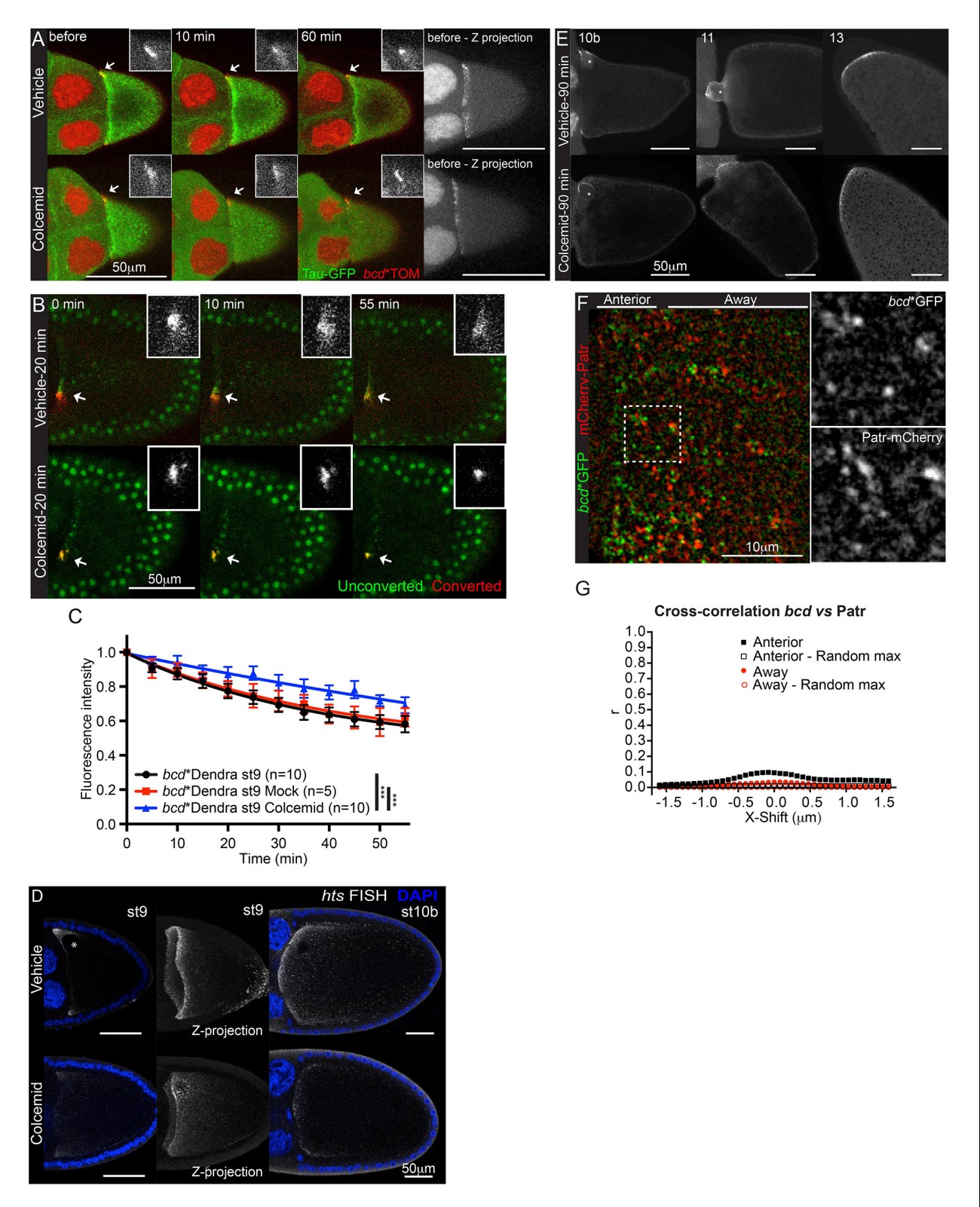

**Figure 4.** *bcd* mRNA is not anchored on microtubules at the anterior of stage 9 oocytes. (A–B) Confocal time-series of stage 9 egg chambers expressing *bcd*\*TOM and Tau-GFP (A) or *bcd*\*Dendra2 (B) treated with the microtubule-depolymerising drug, Colcemid (mock or 400 μg/ml); the arrows indicate anteriorly localised *bcd*\*TOM (A) or photo-converted *bcd*\*Dendra2 (B) and the insets are the corresponding close-ups. (A) Colcemid was added to medium 15 min after the beginning of imaging; Images on the right are maximum intensity projections over the Z-dimension, showing
*Figure 4 continued on next page*

*Figure 4 continued*

the anterior-lateral ring of *bcd*\*TOM. (**B**) Colcemid was added to the medium 20 min prior photo-conversion of localised *bcd*\*Dendra2. (**C**) Graph of the fluorescence decay of photo-converted Dendra2 (**B**), after removal of the fast-recovering, nonspecific component; \*\*\*F-test p value <0.0001. (**D**) Confocal imaging of endogenous *hts* mRNA (FISH) in stage 9 (left - single confocal section; right - maximum intensity projection of the full volume of the oocyte) and stage 10b egg chambers after 90 min treatment with Colcemid (mock or 400 µg/ml); DNA (DAPI) in blue; asterisk indicates the oocyte nucleus. (**E**) Confocal images of *bcd*\*TOM in stage 10b, 11 and 13 egg chambers after 90 min treatment with Colcemid (mock or 400 µg/ml); asterisks indicate the oocyte nucleus. (**F–G**) High magnification wide-field two-colour imaging of *bcd*\*GFP and the minus-end microtubule marker, mCherry-Patronin, in stage 9 oocytes (**F**), and the corresponding Van Steensel co-localisation analysis (**G**).

The following source data and figure supplement are available for figure 4:

**Source data 1.** Van Steensel co-localisation analyses of *bcd*\*GFP and mCherry-Patr.

**Source data 2.** Van Steensel co-localisation analyses of *bcd*\*Tom and Tau-GFP.

**Figure supplement 1.** *bcd*mRNA is not anchored on microtubules at the anterior of stage 9 oocytes.

mRNA was more sensitive than *bcd* mRNA to the depletion of microtubules at stage 9, with only a small amount of RNA remaining localised (***Figure 4D***).

By contrast to stage 9, treating stage 10b egg chambers with Colcemid led to the loss of most *bcd* mRNA from the anterior, except for a small amount around the oocyte nucleus (***Figure 4E***). This is consistent with the lower anterior retention of photo-converted *bcd* mRNA at stage 10b. This reduction in stability coincides with the assembly of new MTOC in the middle of the anterior cortex and the relocalisation of the RNA from an anterior ring to a central disc (***Schnorrer et al., 2000***, ***2002***). The microtubule-dependence of *bcd* mRNA retention is transient, however, and microtubule depolymerisation had little effect on the stability of the localisation at stage 11 and stage 13 (***Figure 4E***).

## *bcd* RNA is not anchored to microtubules

Although Colcemid depolymerises all detectable microtubules in the oocyte, short 'stumps' of microtubules may persist where the minus ends are attached to the cortex, which could provide anchors for *bcd* mRNA. We therefore examined the distribution of *bcd* mRNA particles relative to the microtubule minus ends, using the microtubule minus end-binding protein Patronin as a marker (***Goodwin and Vale, 2010***; ***Hendershott and Vale, 2014***). Although both Patronin-labelled microtubule minus ends and *bcd* mRNA particles are most concentrated at the anterior corners of the oocyte, they very rarely overlap (***Figure 4F***). Analysing their distributions using the van Steensel method gives a Pearson's correlation coefficient of 0.1 at zero displacement, indicating that only a very small proportion of *bcd* mRNA particles co-localise with microtubule minus ends (***Figure 4G***, ***Figure 4—source data 1***) (***van Steensel et al., 1996***). We obtained similar results when using the microtubule-associated protein, Tau, as marker for stable microtubule ends in the presence of Colcemid (***Figure 4—figure supplement 1A–B***, ***Figure 4—source data 2***) (***Parton et al., 2011***). The small degree of co-localisation makes it unlikely that the RNA is anchored to minus ends by static Dynein.

Since microtubules do not appear to anchor *bcd* mRNA, we turned to cortical actin, which has

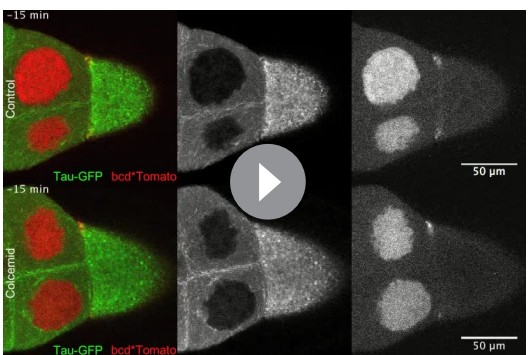

**Video 6.** (related to ***Figure 4A***) – Localised *bcd* mRNA is anchored at the oocyte anterior independently of microtubules. Confocal live imaging of *bcd*\*Tom and the microtubule marker, Tau-GFP, in stage 9 oocytes, with and without depolymerisation of microtubules. Top – mock control; Bottom – Colcemid (400 µg/ml). Colcemid was added 15 min after the start of imaging. Confocal images were acquired every 5 min.

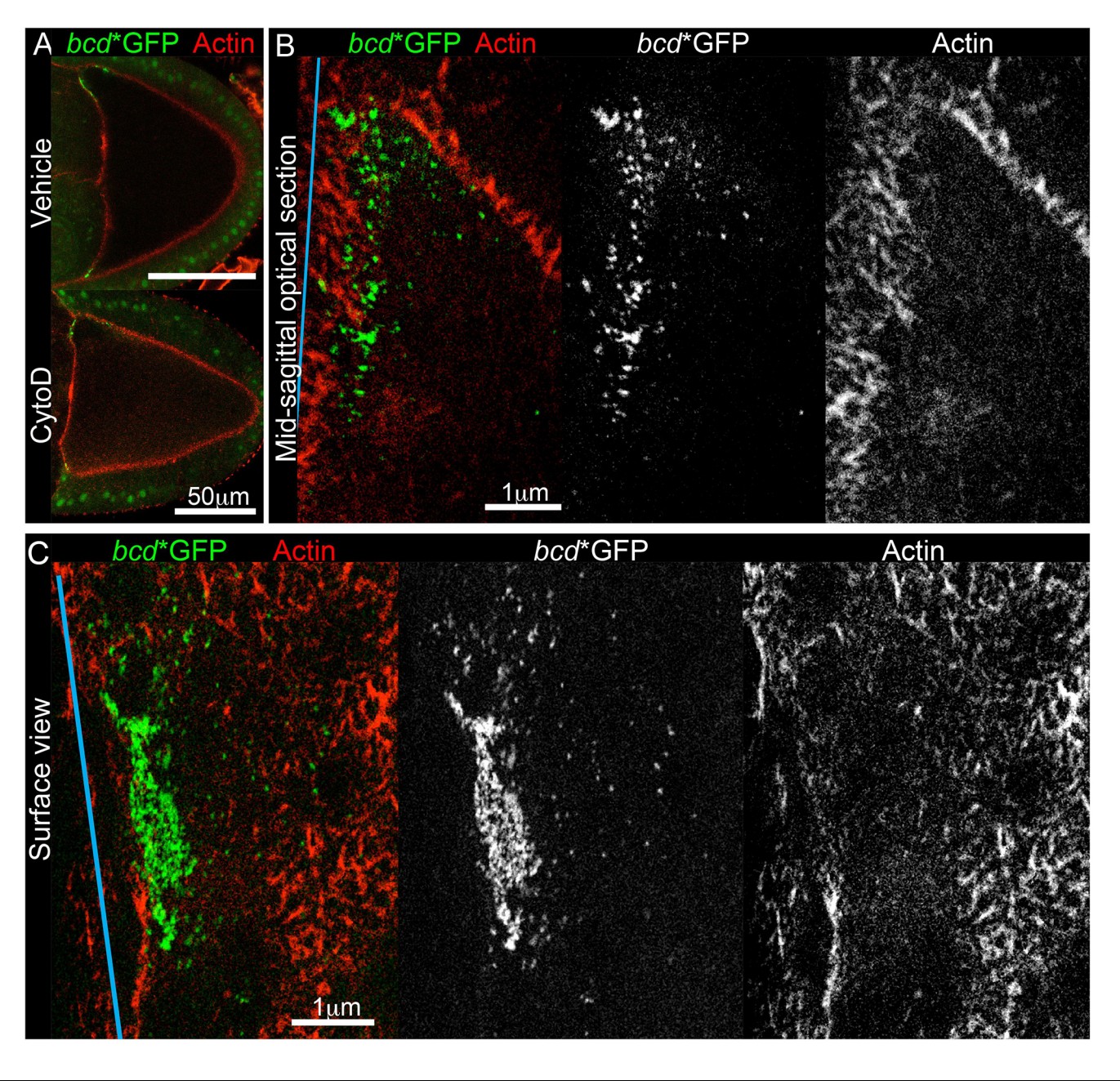

**Figure 5.** *bcd* mRNA is not directly anchored on cortical actin at the anterior of stage 9 oocytes.  (A) Confocal imaging of stage 9 egg chambers expressing *bcd*\*GFP (green) and labelled for actin (Phalloidin-TRITC, red), after 90 min treatment with the actin-depolymerising drug, Cytochalasin D (mock or 10 μg/ml). (B–C) STED super-resolution mid-sagittal (B) or surface (C) images of stage 9 egg chambers expressing *bcd*\*GFP (stained with GFP-Booster-ATTO647N, green) and labelled for actin (Phalloidin-ATTO590, red); spectral unmixing was applied to the images; the blue line indicates the oocyte anterior.

been implicated in keeping the RNA localised at later stages (*Weil et al., 2006*, *2010*). To explore an earlier role of actin on *bcd* mRNA localisation, we cultured stage 9 oocytes in the presence of the actin depolymerising drug, Cytochalasin D. The treatment interfered with the cytoplasmic actin mesh, as it induced premature cytoplasmic streaming (data not shown) (*Dahlgaard et al., 2007*), but did not significantly affect the cortical actin or the distribution of *bcd* mRNA (*Figure 5A*). Since the drug treatment experiment was not conclusive, we applied two-colour Stimulated Emission

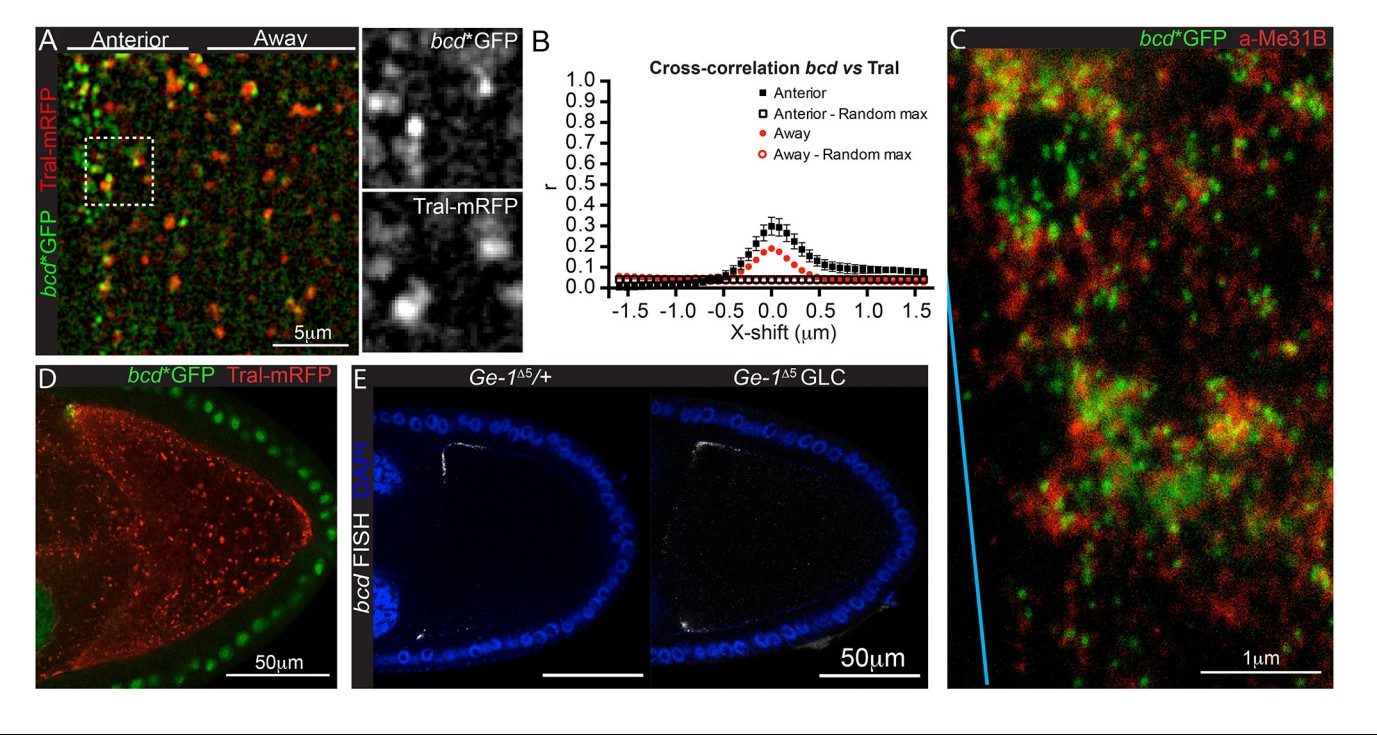

**Figure 6.** *bcd* mRNA partially co-localise to P-bodies at the anterior of stage 9 oocytes. (**A–B**) High magnification wide-field imaging of *bcd*\*GFP (green) and the P-body component, Tral-mRFP (red) (**A**), and the corresponding Van Steensel co-localisation analysis (**B**) in stage 9 oocytes. (**C**) STED super-resolution imaging of a stage 9 egg chamber expressing *bcd*\*GFP (GFP-Booster-ATTO647N, green) and immuno-labelled for the P-body component Me31B (ATTO590, red); spectral unmixing was applied to the image; the blue line indicates the oocyte anterior. (**D**) Confocal image of a wild-type stage 9 oocyte expressing *bcd*\*GFP (green) and Tral-RFP (red). (**E**) Confocal imaging of endogenous *bcd* mRNA (RNA FISH) in wild-type and *Ge-1*$^{\Delta5}$ mutant stage 9 oocytes.

The following source data is available for figure 6:

**Source data 1.** Van Steensel co-localisation analyses of *bcd*\*GFP and Tral-mRFP.

Depletion (STED) super-resolution microscopy to investigate whether *bcd* mRNA is anchored on actin, but detected virtually no co-localisation between them (*Figure 5B–C*). These data suggest that cortical actin is unlikely to act as a direct anchor for *bcd* mRNA during stage 9.

In many cases, RNAs are retained in a specific location by their incorporation into large particles, such as the polar granules at the posterior of the *Drosophila* oocyte or the P granules in *C. elegans* (*Little et al., 2015*; *Trcek et al., 2015*; *Elbaum-Garfinkle et al., 2015*). *bcd* mRNA has been shown to be associated with P-bodies, which may act to prevent its translation during oogenesis (*Weil et al., 2012*). This raises the possibility that sequestration in P-bodies also plays a role in anchoring *bcd* mRNA at the anterior. We observed that *bcd* mRNA particles partially co-localise with P-bodies, particularly at the very anterior of the oocyte. First, van Steensel co-localisation analysis of *bcd* mRNA (*bcd*\*GFP) and the P-body component, Trailerhitch (Tral-mRFP), gave Pearson's correlation coefficients of 0.3 and 0.2 at the very anterior and adjacent cytoplasm of the oocyte, respectively (*Figure 6A–B*, *Figure 6—source data 1*). Second, two-colour STED super-resolution microscopy revealed that *bcd* mRNA particles are enriched within P-bodies, particularly in the larger aggregates, but are also found free in the cytoplasm (*Figure 6C*). P-bodies are ubiquitous in the cytoplasm, however, and are most abundant in the posterior pole plasm (*Figure 6D*). Thus, incorporation of *bcd* RNA into P-bodies could play a role in retaining the RNA at the anterior, but there must be some additional mechanism to ensure that the RNA is only sequestered once it has reached its destination.

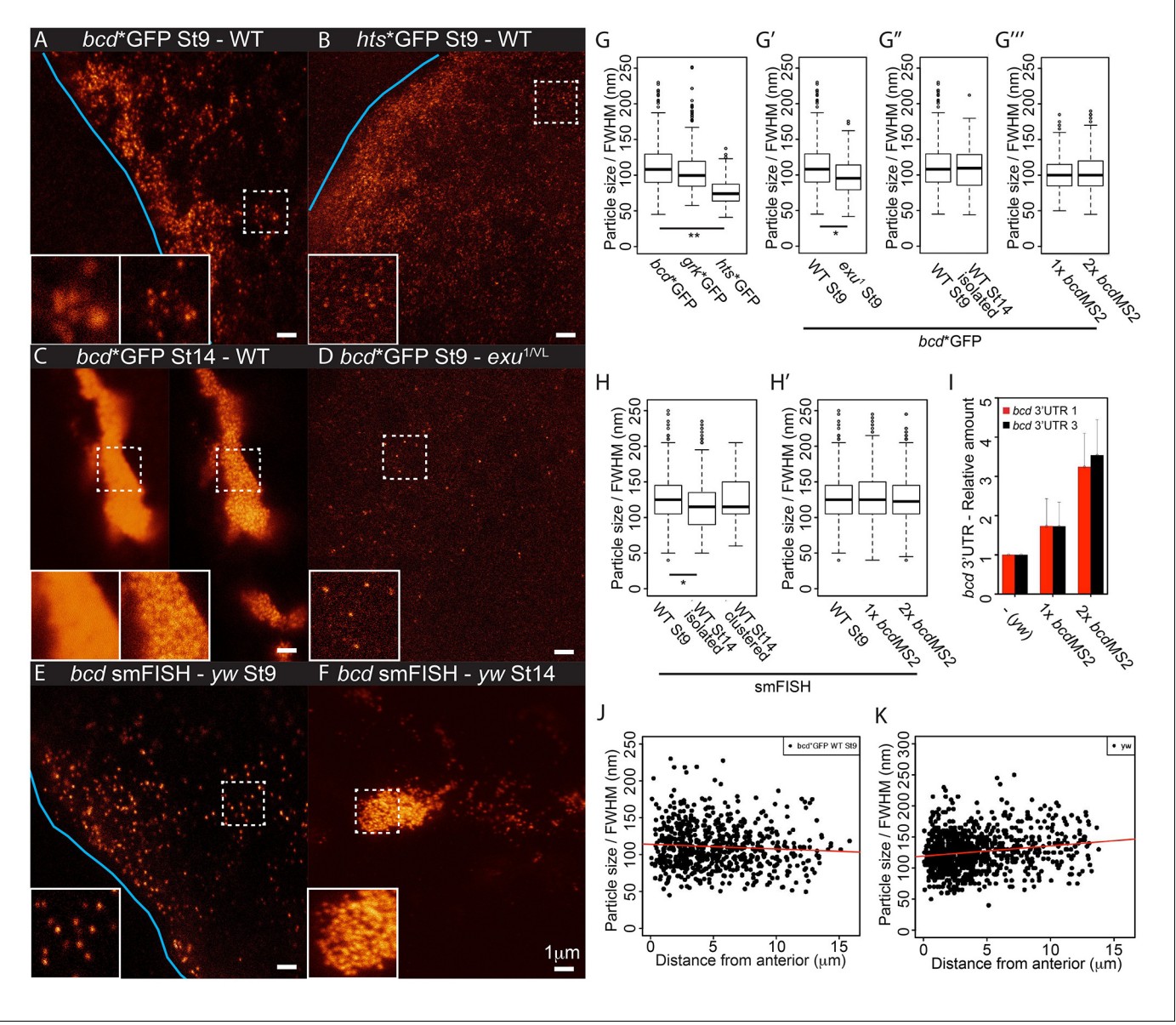

**Figure 7.** *bcd* mRNA assembles into stereotypical particles. (**A–F**) STED super-resolution imaging of mRNA particles labeled with GFP (**A–D**) or single molecule fluorescence *in situ* hybridization (smFISH) (**E–F**). (**A,C,D**) *bcd*\*GFP in wild-type stage 9 (**A**) and stage 14 oocytes (**C**, confocal mode on the left, STED mode on the right), and *exu¹/exuVL* stage 9 oocytes (**D**). (**B**) *hts*\*GFP in wild-type stage 9 oocytes. (**E–F**) smFISH of endogenous *bcd* mRNA in stage 9 and stage 14 oocytes. The blue lines indicate the oocyte anterior; the insets are close-ups of the dashed boxes, confocal mode on the left, STED mode on the right. (**G–H**) Boxplots of the sizes of RNA particles labelled with MCP-GFP (**G**) or smFISH (**H**). (**G**) *bcd*\*GFP, *grk*\*GFP and *hts*\*GFP in stage 9 oocytes. (**G'**) *bcd*\*GFP particles from wild-type and *exu¹/exuVL* stage 9 oocytes. (**G''**) *bcd*\*GFP particles in wild-type oocytes at stage 9 and the isolated particles at stage 14. (**G'''**) *bcd*\*GFP particles in stage 9 oocytes expressing one or two copies of the *bcdMS2* transgene. (**H**) *bcd* RNA particles in wild-type oocytes at stage 9 and isolated or clustered particles at stage 14. (**H'**) Stage 9 oocytes expressing only endogenous *bcd* RNA or one or two additional copies of the *bcdMS2* transgene. (**I**) Relative amounts of *bcd* 3'UTR (RT-qPCR) in ovaries expressing only endogenous *bcd* mRNA (yw) or one or two additional copies of the *bcdMS2* transgene (2 primer pairs, mean ± S.E.M., 3 biological replicates). (**J–K**) Scatterplots of particle sizes versus distance from the anterior in stage 9 oocytes. (**J**) MCP-GFP-labelled transgenic *bcdMS2*. (**K**) smFISH-labelled endogenous *bcd* RNA. \*p<0.05; \*\*p <0.01.

The following figure supplement is available for figure 7:

**Figure supplement 1.** *bcd* mRNA assembles into stereotypical particles.

**Table 5.** Analyses of mRNA particle properties.

| | Genotype | Particles (n) | Particle size ±S.E.M. (nm) | Mixed-effects test P-value a) | Mixed-effects test P-value b) | Summed fluorescence ±S.E.M. (a.u.) | Mixed-effects test P-value c) | Mixed-effects test P-value d) |
|---|---|---|---|---|---|---|---|---|
| MS2-labelling | bcd*GFP - St9 | 732 | 111.9 ± 1.1 | - - - | 0.14 | - - - | - - - | - - - |
| | grk*GFP - St8-9 | 376 | 106.5 ± 1.6 | 0.41 | - - - | - - - | - - - | - - - |
| | hts*GFP - St9 | 284 | 76.6 ± 1.0 | 0.002 ** | 0.56 | - - - | - - - | - - - |
| | bcd*GFP / exu[1] - St9 | 224 | 97.9 ± 1.8 | 0.046 * | - - - | - - - | - - - | - - - |
| | bcd*GFP - St14 (isolated) | 165 | 109.6 ± 2.5 | 0.56 | - - - | - - - | - - - | - - - |
| | bcd*GFP – St9 / 1x bcdMS2 | 293 | 100.2 ± 1.5 | 0.07 | 0.78 | 155.1 ± 5.3 | - - - | 0.11 |
| | bcd*GFP – St9 / 2x bcdMS2 | 292 | 105.3 ± 1.6 | 0.19 | - - - | 149.3 ± 5.1 | 0.50 | - - - |
| smFISH | bcd - St9 endogenous (yw) | 901 | 125.9 ± 1.0 | - - - | <0.0001 *** | 204.4 ± 5.4 | - - - | 0.13 |
| | bcd – St14 (isolated) endogenous (yw) | 749 | 116.3 ± 1.2 | 0.044 * | - - - | 687.7 ± 30.0 | <0.0001 *** | - - - |
| | bcd – St14 (clustered) endogenous (yw) | 125 | 124.6 ± 2.6 | 0.84 | - - - | - - - | - - - | - - - |
| | bcd – St9 endogenous + 1x bcdMS2 | 935 | 128.6 ± 1.0 | 0.28 | - - - | 264.8 ± 7.4 | <0.0001 *** | - - - |
| | bcd – St9 endogenous + 2x bcdMS2 | 1509 | 124.6 ± 0.8 | 0.74 | - - - | 328.5 ± 7.4 | <0.0001 *** | - - - |

a) Mixed effects linear model (LMER) test for comparison of RNA particle sizes (FWHM). Fixed effect: mRNA / Genotype; Random effect: variability between oocytes. Compared to bcd*GFP - St9 or bcd - St9 endogenous (yw)

b) Mixed effects linear model (LMER) test to analyse the effect of the distance from the anterior on the RNA particle sizes (FWHM). Fixed effect: Distance from anterior; Random effect: variability between oocytes.

c) Mixed effects linear model (LMER) test for comparison of the summed fluorescence of RNA particles. Fixed effect: mRNA / Genotype; Random effect: variability between oocytes. Compared to bcd*GFP - St9 / 1x bcdMS2, or bcd - St9 endogenous (yw)

d) Mixed effects linear model (LMER) test for comparison of the summed fluorescence of RNA particles. Fixed effect: Distance from anterior; Random effect: variability between oocytes

- - - Not applicable / Not determined

a.u. arbitrary units

*p<0.05; **p<0.01; ***p<0.001

Source data 1. Properties of RNA particles from STED super-resolution imaging. Includes the data in: *Table 5*; *Figure 7*, panels G–K; *Figure 8*, panles A–D.

To test the role of P-bodies more directly, we examined *bcd* mRNA localisation in germline clones of a null mutation in the core P-body component, *Ge-1*, which has been reported to disrupt P-body structure (*Fan et al., 2011*). The localisation of *bcd* mRNA appeared normal in *Ge-1*[Δ5] clones, however, suggesting that P-body integrity is not required for *bcd* mRNA anchoring (*Figure 6E*).

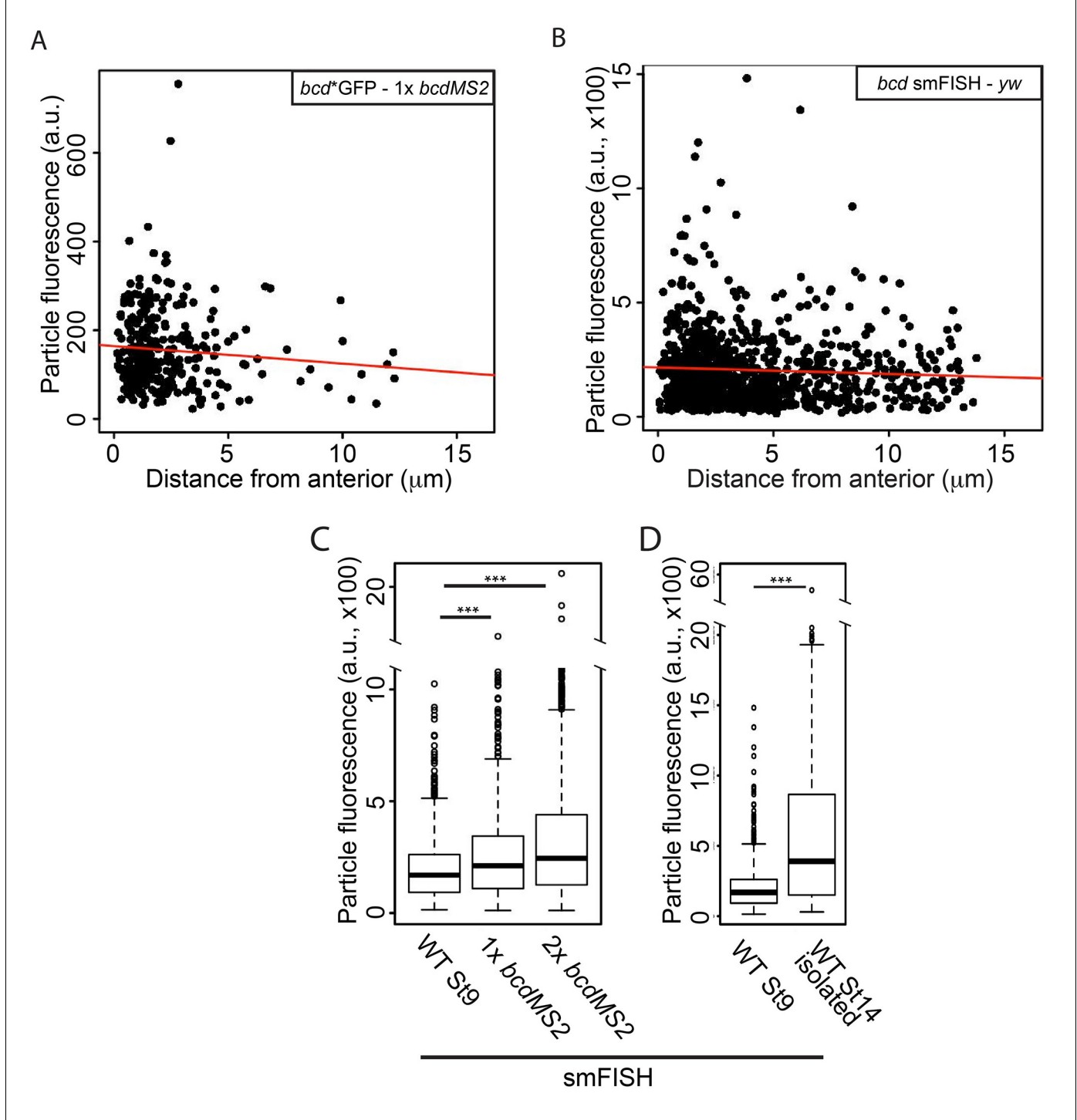

**Figure 8.** The RNA content of the *bcd* mRNA particles increases during oogenesis and with higher gene dosage. (A–B) Scatterplots of the summed fluorescence intensities of *bcd* RNA particles versus distance from the anterior at stage 9. (A) GFP-labelled transgenic *bcdMS2*. (B) smFISH-labelled endogenous *bcd* RNA (*yw* genotype). (C–D) Boxplots of the summed fluorescence intensities of *bcd* RNA particles. (C) smFISH-labelled *bcd* RNA particles from stage 9 oocytes expressing only endogenous *bcd* mRNA (*yw*) or one or two additional copies of the *bcdMS2* transgene. (D) smFISH-labelled *bcd* RNA particles from stage 9 and stage 14 (isolated) oocytes expressing only endogenous *bcd* mRNA (*yw*). ***p<0.001.

## Exu is required for the assembly of *bcd* mRNPs

The highly structured 3'UTR of *bcd* mRNA contains dimerisation/oligomerisation domains (stem-loop III) that are required for its efficient transport and apical localisation in the syncytial blastoderm embryo (*Ferrandon et al., 1997*; *Wagner et al., 2001*; *Snee et al., 2005*). This raises the possibility that the RNA is retained at the anterior by aggregating into larger and less diffusive particles. We therefore used STED microscopy to visualise MS2/MCP-labelled RNA particles with high resolution. Imaging of stage 9 oocytes revealed that *bcd*, *grk* and *hts* mRNAs form regular particles (*Figures 5B-C*,*6C*,*7A–B*, data not shown). We then used a fluorescence intensity curve fitting method to estimate the sizes of the particles (see Material and Methods, *Figure 7—figure supplement 1A*). *bcd* and *grk* mRNAs particles averaged 112 nm and 106 nm, respectively, whereas *hts* mRNA particles were significantly smaller, averaging 77 nm (*Table 5*, *Table 5—source data 1*, *Figure 7A-B,G*).

*bcd* mRNA forms larger aggregates at stage 14 of oogenesis, but super-resolution imaging revealed that these are still composed of small, discrete RNA particles (*Figure 7C*). As the particles in clusters are about 200 nm apart in the XY dimension (224 nm and 202 nm mean distance to nearest neighbour in stage 9 and 14 oocytes, respectively), each ~700 nm optical Z-section is likely to include more than one particle. This causes a high and irregular background, leading to overestimation of fluorescence intensities and unreliable curve fittings. Nevertheless, the estimated size of isolated *bcd* mRNA particles at stage 14 was comparable to those in stage 9 oocytes (*Table 5*, *Table 5—source data 1*, *Figure 7G*). This suggests that *bcd* mRNA particles remain relatively homogeneous in size throughout oogenesis, despite of their clustering into large, semi-ordered aggregates at stage 14.

To confirm these findings, we also performed STED imaging on endogenous *bcd* RNA labelled by single molecule FISH (smFISH) with probes spanning the 3'UTR. This technique also revealed that *bcd* RNA forms particles that remain approximately the same size throughout oogenesis, although they appear slightly larger than those seen with MS2-GFP labelling, presumably because smFISH labels the entire *bcd* 3'UTR, not just the MS2 sites in the RNA (*Table 5*, *Table 5—source data 1*, *Figure 7E–F*). To explore if particle remodelling plays a role in anchoring *bcd* mRNA at the anterior, we compared the properties of particles at different distances from the anterior margin of stage 9 oocytes. The average size of *bcd* mRNA particles measured by both MS2-labelling and smFISH did not change substantially with the distance from the anterior (*Table 5*, *Table 5—source data 1*, *Figure 7J–K*), arguing against their coalescence upon localisation.

The uniform size of *bcd* mRNA particles, regardless of their location or the stage of oogenesis, was unexpected and suggests that the RNA is incorporated into a well-defined structure rather than assembling into aggregates of variable size depending on the RNA concentration. We tested this hypothesis by comparing the sizes of the particles formed by the endogenous RNA in wild-type oocytes with those formed in oocytes expressing either one or two additional copies of *bcdMS2*, which raises the RNA levels to 1.75x and 3.25x the endogenous level, respectively (*Figure 7I*). The size of the *bcd* RNA particles remained constant with increasing RNA concentration (*Table 5*, *Table 5—source data 1*, *Figure 7G–H*), but we observed significantly more particles (172%) in oocytes expressing 2 copies of *bcdMS2* compared to just the endogenous RNA alone. Consistent with this, extra *bcdMS2* RNA did not affect the decay kinetics in photo-conversion experiments, indicating that the diffusion characteristics of the particles were also constant under differing RNA concentrations (*Figure 7—figure supplement 1B*). These experiments demonstrate that the size of *bcd* mRNA particles is insensitive to the concentration of mRNA, supporting the view that the RNA is assembled into a distinct structure of uniform size. Indeed, the only condition that altered the size of the *bcd* mRNA particles was the *exu*[1] mutant, which strongly reduces the affinity of Exu for RNA (*Lazzaretti et al., 2016*). In this case, the few particles that were detected were slightly, but significantly, smaller (98 nm *versus* 112 nm) (*Table 5*, *Table 5—source data 1*, *Figure 7D–E*). Thus, Exu may provide part of the scaffolding for the assembly of *bcd* mRNA particles.

To determine whether the particles also contain the same amount of RNA throughout oogenesis and at different *bcd* gene dosages, we used the summed fluorescent intensities of the particles as a measure of their RNA content. The average fluorescent intensity of the particles detected by both MS2-labelling and smFISH did not change with distance from the anterior, reinforcing the conclusion that the particles do not fuse upon localisation and anchoring at the anterior (*Figure 8A–B*). By

contrast, the mean fluorescence intensity of the particles increased when *bcd* mRNA levels were raised by expressing one or two copies of the *bcdMS2* transgene, with 30% and 60% more fluorescence, and thus more RNA, per particle respectively (*Table 5*, *Table 5—source data 1*, *Figure 8C*). Furthermore, stage 14 particles contained more than 3 times as much RNA as those at stage 9 (*Table 5*, *Table 5—source data 1*, *Figure 8D*). It has previously been shown that most *bcd* RNA enters the oocyte during nurse cell dumping at stage 10b-12 (*Weil et al., 2006*), suggesting that much of this extra RNA is incorporated into pre-existing RNA particles. Thus, the *bcd* RNA particles have a variable RNA content, despite their constant size.

## Discussion

It has been generally assumed that *bcd* mRNA is localised by directed transport along a polarised microtubule cytoskeleton (*Wolpert et al., 2015*; *St Johnston, 2005*). Indeed, studies on the later stages of oogenesis indicate that the RNA is continually transported towards microtubule minus ends by Dynein and then is increasingly more anchored as oogenesis progresses (*Pokrywka and Stephenson 1991*; *Weil et al., 2006*, *2008*). Here we provide evidence that the RNA is localised by a different mechanism at other stages.

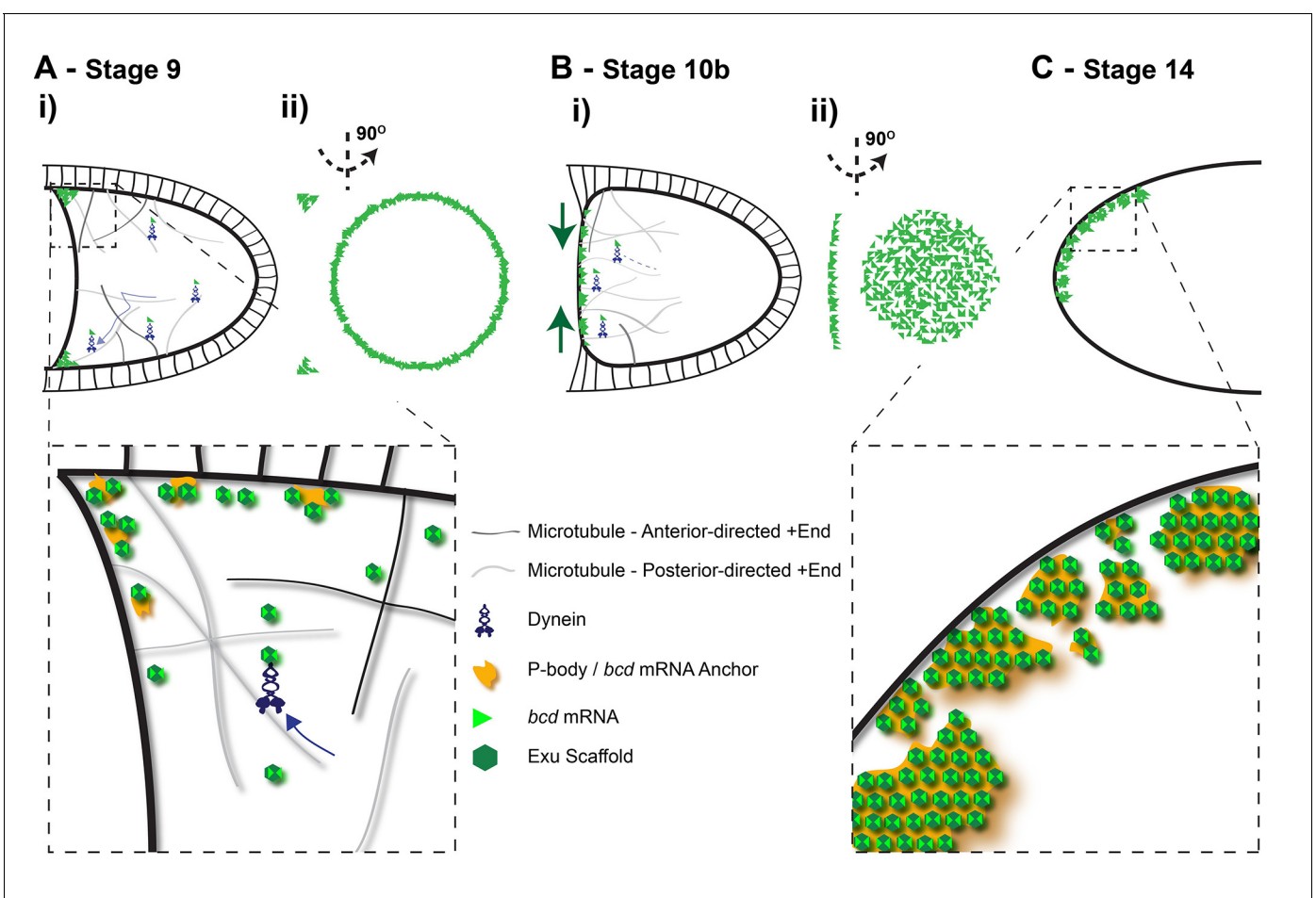

**Figure 9.** Diagram of the steps in *bcd* mRNA localisation during oogenesis. (**A**) Stage 9 of oogenesis: *bcd* mRNA localises to the anterior-lateral margins of the oocyte (i), forming a ring when viewed end on (ii). Close-up: *bcd* mRNA is assembled into Exu-dependent particles that are actively transported by Dynein along an unpolarised microtubule cytoskeleton; On reaching the anterior, *bcd* RNA particles are anchored independently of microtubules, possibly by docking to P-bodies. (**B**) Stage 10b of oogenesis: Following the reorganisation of microtubule minus-ends, *bcd* mRNA re-localises from the anterior-lateral margin to form a disc at the centre of the anterior cortex of the oocyte. (**C**) Stage 14 of oogenesis: *bcd* mRNA particles cluster into large aggregates at the oocyte cortex.

Live imaging of fluorescently-labelled *bcd* mRNA in stage 9 oocytes revealed that the RNA forms particles that undergo frequent active movements along microtubules. The speed was strongly reduced in *Dhc* mutants, consistent with *bcd* mRNA being transported predominantly by Dynein. Furthermore, *bcd* mRNA moved significantly more frequently and faster in a null mutant in *Khc*. Since Kinesin-I transports Dynein to the oocyte posterior, the more frequent movements in the *Khc* null mutant may be explained by the higher levels of Dynein at the anterior of the oocyte, which should increase the probability of its binding to *bcd* RNPs. This cannot account for the increased velocity of *bcd* mRNA particle movements, however, as this also occurs in the "slow" *Khc* alleles, which still localise Dynein posteriorly. Thus, Kinesin-I somehow slows down Dynein through a mechanism that depends on its full motor activity. One possibility is that Kinesin-I engages in a tug of war with Dynein and therefore exerts a drag that slows down Dynein movement. Interestingly, inhibition of Dynein increases both the velocity of Kinesin-I-driven ooplasmic streaming and *osk* mRNA particle transport, indicating that this antagonistic relationship between Dynein and Kinesin-I is reciprocal (*Serbus et al., 2005*; *Zimyanin et al., 2008*).

*osk* mRNA is transported to the posterior of the oocyte by Kinesin-I-dependent movements along a weakly polarised microtubule network, in which approximately 15% more of the microtubules have their plus-ends pointing posteriorly than anteriorly (*Zimyanin et al., 2008*; *Parton et al., 2011*). We found that *bcd* mRNA particles show a reciprocal anterior bias near the middle of the oocyte, with 12% more movements towards the anterior cortex than away from it. This supports the idea that *bcd* mRNA is mainly transported towards microtubule minus ends by Dynein. The bias becomes increasingly weak close to the anterior, however, and even reverses in the region 0-5 μm from the anterior cortex. Since the small bias in the anterior region is unaffected in the *Khc* null mutant and there are virtually no directional reversals, it is not due to bidirectional transport along a polarised microtubule network. Instead, there seem to be approximately equal numbers of microtubules pointing anteriorly and posteriorly near the anterior, with most microtubules running parallel to the anterior cortex. This fits well with experimental measurements of microtubule polarity in the oocyte, in which the orientation bias decreases from posterior to anterior, and with 3-dimensional computer simulations of the microtubule organisation, which give negligible orientation bias close to the anterior (*Parton et al., 2011*; *Trong et al., 2015*). Thus, directed transport cannot account for the localisation of *bcd* mRNA to the very anterior of the oocyte, although it can deliver the mRNA to a broader anterior region.

In light of these observations, we propose that *bcd* mRNA is localised by rapid, bidirectional Dynein-dependent transport in and out of the anterior region, coupled to some mechanism that specifically retains or anchors the RNA at the anterior cortex (*Figure 9A*). This random transport and anterior anchoring model predicts that the RNA will only turn over slowly at the anterior cortex. This is confirmed by FRAP and photo-conversion experiments, which show that more than 60% of the RNA remains stably localised at the anterior over a period of 55 min. This turn-over rate is the same as that measured for *grk* mRNA, which has previously been shown to be specifically anchored at its localisation site above the oocyte nucleus (*Delanoue et al., 2007*; *Jaramillo et al., 2008*). Further support for non-directional transport and anterior anchoring comes from the observation that *bcd* mRNA is more efficiently retained at the anterior when the microtubules are depolymerised, indicating that microtubule-based transport plays a role in removing the RNA from the anterior, as well as delivering it.

Unlike *bcd* mRNA, the behaviour of *hts* mRNA at stage 9 fits well with the predictions of the continual active transport model. It is localised to a broader anterior region than *bcd* mRNA, turns-over significantly more rapidly than *bcd* RNA in photo-conversion experiments and spreads along the anterior margin after photoconversion. Furthermore, *hts* mRNA localisation is strongly reduced after 90 min of Colcemid treatment and in the *shot*[2A2] mutant, in which the oocyte microtubule cytoskeleton is not polarised, whereas *bcd* mRNA localisation is largely unaffected in both conditions. Thus, the contrast between the two RNAs reinforces the view that *bcd* mRNA cannot be explained solely by directed transport and must involve an anterior anchoring step.

This model may help to explain the observations of *Cha et al. (2001)*, who showed that *bcd* mRNA injected into the oocyte localises to the nearest region of anterior/lateral cortex, whereas RNA that is exposed to nurse cell cytoplasm before injection localises specifically to the anterior. Although the authors proposed that the "nurse cell-conditioned" RNA gains the capacity to discriminate between microtubules emanating from the anterior and lateral cortex, a simpler explanation is

that both untreated and conditioned RNA are transported by Dynein along microtubules, but only the latter becomes competent to be retained at the anterior. The untreated RNA therefore concentrates near microtubule minus ends, much like *hts* RNA (which is more biased towards the anterior than injected RNA because it enters from the nurse cells), whereas the conditioned RNA localises specifically to the anterior. Thus, factors such as Exu loaded on the RNA in the nurse cells may licence the RNA for anterior anchoring (*Figure 9A*).

The retention of *bcd* mRNA at the anterior varies over the course of oogenesis, with the RNA being much less stably localised at stage 10b. This fits well with the observations of *Weil et al., 2006*, who measured very similar fluorescence recovery rates at stage 10b to those reported here. This decrease in anterior retention coincides with a redistribution of the RNA from an anterior ring to a disc at the centre of the anterior cortex, and with the formation of a new MTOC in this region (*St Johnston et al., 1989*; *Schnorrer et al., 2002*; *Vogt et al., 2006*). Thus, the anterior anchoring mechanism seems to be specifically inactivated at this stage to allow the remodelling of the RNA distribution (*Figure 9B*). During this period, *bcd* mRNA localisation is consistent with continual active transport along the polarised microtubule network formed by the new anterior MTOC. This is only transient, however, as the RNA becomes more stable at the anterior at stage 13, and is very efficiently anchored at stage 14, which is important to keep *bcd* mRNA localised until fertilisation, so that it can act as the source of the Bcd morphogen gradient in the embryo (*Figure 9C*).

The mechanism that retains *bcd* RNA at the anterior is unclear. We can rule out anchoring by Dynein to microtubule minus ends, as has been reported for *grk* mRNA in the oocyte and pair-rule transcripts in the embryo (*Delanoue and Davis, 2005*; *Delanoue et al., 2007*), since the anterior retention of *bcd* mRNA is not microtubule-dependent and the RNA does not co-localise with microtubules. The mRNA could be tethered to cortical actin, which would be consistent with the anchoring defect in late oocytes seen in *swallow* mutants, which disrupt the actin cortex (*Weil et al., 2010*). However, *bcd* RNA does not show a significant co-localisation with F-actin, although this tethering could be indirect. Another possibility is that the RNA is maintained at the anterior by sequestering it in P-bodies (*Weil et al., 2012*) (*Figure 9*). P-bodies are ubiquitous throughout the oocyte, and there would therefore have to be some mechanism that induces *bcd* RNA incorporation into these structures specifically at the anterior.

Super-resolution imaging revealed that *bcd* mRNA forms 110–120 nm particles throughout oogenesis, regardless of whether the RNA is localised or not. Even the large aggregates of RNA at stage 14 are still formed of individual particles of similar size, although their protein composition is different from stage 9, as Staufen and ESCRT-II are recruited to *bcd* mRNA only at stage 10b (*Martin et al., 2003*; *Weil et al., 2006*) (*Figure 9*). Importantly, over-expression of the mRNA does not alter particle size, but instead leads to more particles, which have higher average RNA content. The same occurs at stage 14 of oogenesis, when the *bcd* mRNA content of the oocyte is much higher following nurse cell dumping. Thus, *bcd* mRNA seems to assemble into a structure of defined size, almost like a virus particle, which can incorporate more or less mRNA molecules depending on availability. An *exu* mutant that affects RNA binding affinity causes a large reduction in the number of detectable *bcd* mRNA particles and a small, but significant reduction in the size of the few particles that form. This suggests that Exu, which forms homodimers that probably bind two *bcd* mRNA molecules (*Lazzaretti et al., 2016*), plays a role in scaffolding the assembly of the particles (*Figure 9*). Loss of Exu also strongly reduces both the speed and frequency of *bcd* mRNP movement, as well as its anterior anchoring at all stages of oogenesis. Particle formation may therefore be a prerequisite for all of these processes, explaining the pleiotropic effects of *exu* mutants.

The invariant size of *bcd* RNA particles make them fundamentally different from other well-characterised RNA granules, such as the P-granules in *C. elegans,* which form by the aggregation of RNA and proteins into droplets that phase-separate from the surrounding cytoplasm (*Brangwynne et al., 2009*; *Saha et al., 2016*). P-granules have variable size that depends on the RNA concentration and readily fuse with each other when juxtaposed. By contrast, *bcd* RNA particles stay the same size as the RNA concentration increases, even though they incorporate more RNA, and they do not appear to fuse when tightly clustered in aggregates. The exact nature of the particles will require the identification of more of their components, but their behaviour is compatible with a model in which they consist of a rigid protein framework that contains multiple RNA-binding sites. In future, it will be interesting to determine whether other localised RNAs are packaged into similar structures.

## Materials and methods

### *Drosophila* stocks and genetics

The *bcdMS2* transgene was generated by inserting 11 MS2-binding sites (C-loop form) (*Lowary and Uhlenbeck, 1987*) into the *SpeI* restriction site at the 5′-terminus of *bcd* 3′UTR (FlyBase ID: FBgn0000166), which was cloned downstream of the maternal α4 tubulin promoter. The *htsMS2* transgene was generated by cloning the cDNA of the N4 isoform of *hts* (Flybase ID: FBgn0263391; *Whittaker et al., 1999*), excluding the 5′UTR and start codon, downstream of the maternal α4 tubulin promoter and inserting 10 MS2-binding sites between the *SpeI* and *NotI* restriction sites at the 5′-terminus of the 3′UTR. The hsp83-MCP-Dendra2 transgene is identical to the hsp83-MCP-GFP transgene (*Forrest and Gavis, 2003*) except that the EGFP sequences are replaced by Dendra2 sequences (Evrogen, Russia). The osk-NLS-MCP-Tomato transgene was generated by inserting the cDNA of tdTomato (*Shaner et al., 2004*) after NLS-MCP, which was cloned from hsp83-NLS-MCP-GFP (*Forrest and Gavis, 2003*). For germline-specific expression, the NLS-MCP-Tomato fusion was cloned downstream of the *osk* promoter, cloned from an *osk* rescue construct (gift from Anne Ephrussi).

The *mRNA-MS2* fusion transgenes were recombined with hsp83-NLS-MCP-GFP, osk-NLS-MCP-Tomato or hsp83-NLS-MCP-Dendra2. Germline clones (GLC) were generated using the ovoD/FLP system by heat-shocking second to third instar larvae for 2 hr at 37°C for 3 consecutive days (*Chou and Perrimon 1996*).

Other fly strains used were:
$y^1w^1$ (Bloomington stock 1495);
osk-$(MS2)_{10}$ (*Zimyanin et al., 2008*);
$shot^{2A2}$ (*Chang et al., 2011*);
cn, $exu^1$, bw ([*Schüpbach and Wieschaus, 1986*], Bloomington stock 1989);
cn, $exu^{VL}$, bw (*Hazelrigg et al., 1990*);
FRT42B, c, $Khc^{27}$ (*Brendza et al., 2000*);
FRT42B, c, $Khc^{17}$ (*Brendza et al., 2000*);
FRT42B, c, $Khc^{23}$(*Brendza et al., 2000*);
$Dhc64C^{6-10}$ ([*McGrail and Hays, 1997*], Bloomington stock 8747);
$Dhc64C^{8-1}$ (*Gepner et al., 1996*);
$Dhc64C^{6.10}$, FRT2A (Gift from U. Abdu);
Tau-GFP (*Micklem et al., 1997*);
grk-$(MS2)_{12}$, MCP-GFP (grk*GFP) (*Jaramillo et al., 2008*);
Ubq-Dlic-GFP (*Baumbach et al., 2015*);
$grk^{2B6}$, $grk^{2E12}$ (*Neuman-Silberberg and Schüpbach, 1993*);
UAS:mCherry-Patr (*Nashchekin et al., 2016*);
nanos:GAL4-VP16 ([*Van Doren et al., 1998*], Bloomington stock 64277);
Tral-mRFP trap line (*Lowe et al., 2014*);
Me31B-GFP trap-line (*Buszczak et al., 2007*);
FRT2A, Ge-$1^{Δ5}$ (*Fan et al., 2011*).

### Immunological/staining methods

Immunofluorescence

Ovaries from 48–72 hr old females were dissected in PBS-T (PBS + 0.2% Tween-20) and fixed in 4% formaldehyde in PBS-T for 20 min. For analyses of stage 14 oocytes, ovaries from 72–96 hr old virgin females were dissected into Modified Robb's medium and fixed in Cacodylate Fixative for 20 min (*McKim et al., 2009*).

The fixed samples were then incubated with 5% BSA in PBS-T for 1 hr to block nonspecific antigen-antibody reactions, incubated with primary antibodies in PBS-T plus 1% BSA at 4°C for 18 hr and then washed in PBS-T. If the primary antibody was not directly conjugated with a fluorophore, ovaries were further incubated with fluorophore-conjugated secondary antibodies (for confocal microscopy – 1:200, Jackson ImmunoResearch laboratories, PA, USA; for STED microscopy – ATTO590-labelled anti-mouse antibody, 1:200, Enzo Life Sciences, UK) and then washed in PBS-T,

before addition of Vectashield mounting medium (Vector Laboratories, CA, USA, Cat# H-1000 RRID: AB_2336789).

The primary antibodies used were: anti-Me31B mouse monoclonal antibody (1:1000) (*Nakamura et al., 2001*) (Cat# me31B RRID:AB_2568986), FITC-conjugated anti-α-Tubulin mouse monoclonal antibody (1:100, Sigma-Aldrich, MO, USA, Cat# F2168 RRID:AB_476967) and GFP-Boosters (GFP-nanobody directly conjugated to Abberior STAR RED or ATTO 647N, 1:400, Chromotek, Germany, Cat# gba647n-100, RRID:AB_2629215). F-actin was labelled by incubating ovaries in PBS-T with Phalloidin conjugated to either TRITC (1:500, Sigma-Aldrich, MO, USA) or ATTO590 (1:2000, ATTO-TEC, Germany).

### Conventional fluorescence *in situ* hybridisation (FISH)

Fluorescence *in situ* hybridisations (FISH) were performed according to standard protocols. The anti-sense probes for *bcd* and *hts* RNAs were synthesised using the DIG RNA Labelling mix (Roche, Switzerland) and the linearised plasmids: pGEM_bcd (*Driever et al., 1990*) (cut with *BamHI*) and pNB40_htsN4 (*Whittaker et al., 1999*) (cut with *SalI*).

### Single molecule FISH (smFISH)

Ovaries from 48–72 hr old females were dissected in PBS and fixed in 4% formaldehyde in PBS for 20 min. For stage 14 oocytes, ovaries from 72–96 hr old virgin females were dissected into Modified Robb's medium, fixed in Cacodylate Fixative for 20 min (*McKim et al., 2009*) and washed in PBS. Samples were dehydrated in ethanol at 4°C for 18 hr, incubated in wash buffer (2X saline-sodium cit-rate (SSC), 10% formamide) for 30 min, and then hybridized with 500 nM ATTO647N-conjugated antisense Stellaris probes for *bcd* RNA (Biosearch Technology, UK) in hybridization buffer (10% dex-tran sulfate, 2X SSC, 10% formamide) at 37°C for 4 hr. After 2 hr in wash buffer, samples were mounted in Vectashield mounting medium (Vector Laboratories, CA, USA, Cat# H-1000 RRID:AB_2336789).

Antisense RNA probes targeting the 3'UTR of *bcd* RNA:

5'-GAAACTCTCTAACACGCCTC-3', 5'-ACAGTGGTTAACCTAAAGCT-3', 5'-TGGTATTTGTACAA TCAGGA-3', 5'-CTTTCTACGCGTAGATATCT-3', 5'-ACGGATCTTAGGACTAGACC-3', 5'-AAAC TTCCCTGGGAACCATT-3', 5'-CTGCTGACTAGGCTAGTACA-3', 5'-GATATGCACTGGAATCCGTG-3', 5'-GAGTTAACTGGAGTATCACT-3', 5'-AGCGTATTGCAGGGAAAGTA-3', 5'-CACCCAGATACA TCTAAGGC-3', 5'-CATATTCCCGGGCTTTAGTG-3', 5'-TGGCCTCAAATGTAACTGGT-3', 5'-AC TTTCCATGGAATACGCTT-3', 5'-ATTTCCGAAATGTGGGACGA-3', 5'-AGAAGATTTCTTGCTGGC T-3', 5'-GTACAGTTTTTAGCTATGTC-3', 5'-ATGAGATTACGCCCAAGAGA-3', 5'-ATGTTCGATC TTTAAGGGTA-3', 5'-ACACTTTGGCATAGCATAGA-3', 5'-GCGCAAATGTTTGATTATGT-3', 5'-TTGC TGACTATTCTTGGTCA-3', 5'-ACAAATGGTCTGCATTGATT-3', 5'-TGATAGTTATTCCGTTTGGC-3', 5'-ATGCTCTTCTTAGTGATGTA-3', 5'-ACTTGAGGCCTAACAGATTG-3', 5'-ACAACATCAAAGG TGCAGCA-3' & 5'-ATTTACCCGAGTAGAGTAGT-3'.

## Drug treatments

Microtubules were depolymerised using Colcemid (Sigma-Aldrich, MO, USA). For acute depolymer-isation of microtubules, ovaries from 48–72 hr old females were dissected in live imaging medium (5 µg/ml insulin and 2.5% foetal calf serum in Schneider's medium (Sigma-Aldrich, MO, USA; adapted from *Bianco et al., 2007*), in a Poly-L-Lysine-coated imaging chamber (Thistle Scientific, UK). Colcemid was then added to 400 µg/ml. The colocalisation between *bcd* mRNA and microtubule minus ends was examined in flies expressing *bcd*\*Tomato and Tau-GFP that were starved for 2 hr and then fed fresh yeast paste containing 100 µg/ml Colcemid for 2 hr (*Pokrywka and Stephenson, 1995*). We depolymerised F-actin by dissecting ovaries in live imaging medium and then adding Cytochalasin D to 10 µg/ml (Sigma-Aldrich, MO, USA; *Emmons et al., 1995*).

## Confocal imaging

Confocal imaging was performed on an Olympus IX81 FV1000 laser scanning confocal microscope (Olympus, Japan) using 40x UPlanFLN 1.3NA or 60x UPlanSApo 1.35NA oil immersion objectives

(Olympus, Japan) and the Olympus Fluoview FV10-ASW software (Olympus, Japan, RRID:SCR_014215).

## Live imaging for particle tracking and co-localisation analyses

Ovaries from 48–72 hr old females were dissected directly onto coverslips in 10S Voltalef oil (VWR International, PA, USA). For acute drug treatments, drugs were added for 20 min to ovaries in live imaging medium (see above) in Poly-L-Lysine-coated imaging chambers (Thistle Scientific, UK). Ovaries were transferred onto coverslips and finely dissected in 10S Voltalef oil. Imaging was performed on either a wide field DeltaVision microscope (Applied Precision, WA, USA) equipped with a Photometrics 512 EMCCD camera (Photometrics, AZ, USA) and a 2x magnification tube fitted between the unit and the camera, or on an Olympus IX81 inverted microscope (Olympus, Japan) combined with a Yokogawa CSU22 spinning disk confocal imaging system and an iXon DV855 camera (ANDOR Technology, UK). The softWorXs software (Applied Precision, WA, USA) was used to acquire and deconvolve images on the DeltaVision system and MetaMorph Microscopy Automation and Image Analysis Software (Molecular Devices, CA, USA, RRID:SCR_002368) was used to acquire images on the spinning-disk microscope. A 100x UPlanSApo 1.4 NA oil immersion objective lens (Olympus, Japan) was used in both systems.

### Particle tracking

Moving particles were tracked manually using the MTrackJ plugin for the Fiji image analysis software (Fiji, RRID:SCR_002285) (*Meijering et al., 2012*; *Schindelin et al., 2012*). We analysed at least 4 oocytes per sample type, and tracked all visible moving particles in each movie. The speed, distance and directionality of each moving particle were calculated with Excel software (Microsoft, CA, USA). For each moving particle, the speed was calculated as the mean of its velocities at each individual timepoint, the distance from the anterior was measured from its initial position and the direction of movement was defined by the vector between the initial and final positions.

Particle speeds in different samples were compared using the Wilcoxon rank-sum test (univariate analyses) or a mixed-effects linear model (multivariate analyses; fixed effect variable – genotype; random effect variables – oocyte and movie). The binomial test was used to test whether the anterior directional bias was significantly larger than zero. To test whether the net anterior displacement of particles was significantly larger than zero we performed the Wilcoxon 1-sample test.

The mobile fraction of mRNA particles was calculated as the proportion of particles that undergo active movements in 5 s periods. We excluded the very anterior of the oocyte from these measurements because the *bcd* mRNA particles were at too high a density to distinguish individual particles. The mobile fractions of different samples were compared using the T-test. Statistical tests were performed on the software R (R Project for Statistical Computing, RRID:SCR_001905) (*R Core Team, 2013*).

### Co-localisation analyses

The background was subtracted from the two-colour images using an 8 pixel rolling-ball filter. For the co-localisation analysis, we used ESCoP, a plugin for Fiji that implements a combination of Van-Steensel's cross-correlation function (*van Steensel et al., 1996*) and Costes' randomisation (*Costes et al., 2004*).

## FRAP and Photo-conversion analyses

Ovaries from 48–72 hr old females were dissected directly onto coverslips in 10S Voltalef oil (VWR International, PA, USA), except when treated with drugs, in which case the dissections were performed in live imaging medium (see above) in a Poly-L-Lysine-coated imaging chamber (Thistle Scientific, UK); drugs were added to the medium 20 min before imaging. Fluorescence recovery after photobleaching (FRAP) and photo-conversion experiments were performed on an Olympus IX81 FV1000 laser scanning confocal microscope, (Olympus, Japan) equipped with the Olympus Fluoview FV10-ASW software (Olympus, Japan, RRID:SCR_014215) and either a 60x UPlanSApo 1.35 NA oil immersion objective (Olympus, Japan; for dissections in oil) or a 60x UPlanSApo 1.2 NA water immersion objective (Olympus, Japan; for dissections in live imaging medium). All imaging

conditions (laser power, bleached or photo-converted area, image dimensions, pixel scanning time and time points) were kept constant in all samples. At least 5 oocytes were analysed per sample type.

## Curve fitting

Mean fluorescence intensities of photobleached (FRAP) or photo-converted regions of interest (ROIs) were measured manually on Fiji (RRID:SCR_002285). Curve fitting was then performed by non-linear least-squares fitting using the statistical software, GraphPad Prism 6 (Graphpad Software, CA, USA, RRID:SCR_002798).

## Normalisation for photobleaching during image acquisition

After substraction of the background, the fluorescence intensities of the photobleached or photo-converted ROIs were normalised for photobleaching during image acquistion, which was calculated as the fluorescence decay of MCP-GFP (FRAP) and of photo-converted MCP-Dendra2 (Photo-conversion) in fixed egg chambers. These measurements were fitted to single exponential equations of the form:

$$I(t) = I_0 * e^{-(p*t)} \tag{1}$$

where $I(t)$ is the fluorescence intensity as a function of time $(t)$, $I_0$ is the initial fluorescence intensity, and $p$ is the rate of photobleaching (*Vicente et al., 2007*). The normalised datasets were then obtained by the following equation:

$$NI(t) = I_t / e^{-(p*t)} \tag{2}$$

where $NI(t)$ is the normalised fluorescence intensity as a function of time $(t)$, $I_t$ is the measured fluorescence intensity at $t$ time, and $p$ is the rate of photobleaching during imaging acquisition determined by *Equation 1*.

## FRAP

FRAP data from samples with a single component/behaviour follow a single exponential equation of the form:

$$NI(t) = 1 - F_{IM} - F_{MOB} * e^{-(\ln(2)*t/\tau)} \tag{3}$$

where $NI(t)$ is the normalised fluorescence intensity as a function of time $(t)$, $F_{IM}$ is the immobile fraction, $F_{MOB}$ is the mobile fraction, and $\tau$ is the recovery half-time of the mobile fraction (*Bulinski et al., 2001*; *Sprague et al., 2004*; *Bulgakova et al., 2013*). However, the fluorescence recovery of localised *bcd*\*GFP in stage 9 oocytes was better fit by a bi-exponential, with fast and slow recovering populations (*Figure 3A,D*). Because microtubule depolymerisation only affected the slow-recovering component of the signal and we observed very fast fluorescence recovery at the anterior of oocytes that only expressed MCP-GFP and in the nurse cell cytoplasm of *bcd*\*GFP egg-chambers (*Figure 3B–D*), the fast component is likely to be nonspecific signal from autofluorescence and/or free MCP-GFP. We therefore fitted the FRAP data from nurse cell cytoplasm to *Equation 3*, in order to determine the recovery half-time of the nonspecific fast recovering signal (2.0 min). With this parameter, we were then able to better fit the localised RNA FRAP data to a bi-exponential composed of both the nonspecific signal and the RNA signal:

$$NI(t) = 1 - F_{IM} - C_{NS} * e^{-(\ln(2)*t/\tau_{NS})} - C_{RNA} * e^{-(\ln(2)*t/\tau_{RNA})} \tag{4}$$

where $NI(t)$ is the normalised fluorescence intensity as a function of time $(t)$, $F_{IM}$ is the immobile fraction, $\tau_{NS}$ (2.0 min) and $\tau_{RNA}$ are the decay half-times of the nonspecific and RNA signals, and $C_{NS}$ and $C_{RNA}$ are the respective mobile fractions. This allowed us to remove the nonspecific component and fit the remaining signal to a single exponential (*Equation 3*) that describes the behaviour of only the RNA.

## Photo-conversion

Single component/behaviour photo-conversion data also follows a single exponential equation, but of the form:

$$NI(t) = F_{IM} + F_{MOB} * e^{-(\ln(2)*t/\tau)} \tag{5}$$

where $NI(t)$ is the normalised fluorescence intensity as a function of time $(t)$, $F_{IM}$ is the immobile fraction, $F_{MOB}$ is the mobile fraction, and $\tau$ is the half-time of the mobile fraction.

Like the fluorescence recovery in the FRAP experiments, photo-conversion of Dendra2-labelled *bcd* mRNA (*bcd*\*Dendra2) at the anterior of stage 9 oocytes yielded a two-phase decay, consistent with the existence of a rapidly-diffusing nonspecific signal and a slower specific signal. The signal obtained by photo-conversion at the anterior of oocytes expressing only MCP-Dendra2 or in nurse cells of *bcd*\*Dendra2-expressing egg chambers decayed very rapidly (*Figure 3—figure supplement 1A-B*). We therefore fitted the photo-conversion data from the nurse cell cytoplasm to *Equation 5*, in order to determine the decay half-time of the nonspecific signal (3.2 min). With this parameter, we were then able to better fit the photo-conversion data of localised RNA to a bi-exponential that included both the nonspecific signal and the RNA signal:

$$NI(t) = F_{IM} + C_{NS} * e^{-(\ln(2)*t/\tau_{NS})} + C_{RNA} * e^{-(\ln(2)*t/\tau_{RNA})} \tag{6}$$

where $NI(t)$ is the normalised fluorescence intensity as a function of time $(t)$, $F_{IM}$ is the immobile fraction, $\tau_{NS}$ (3.2 min) and $\tau_{RNA}$ are the decay half-times of the nonspecific and RNA signals, and $C_{NS}$ and $C_{RNA}$ are the respective mobile fractions. This allowed us to subtract the nonspecific component and fit the remaining signal to a single exponential (*Equation 5*) that describes the behaviour only of the RNA.

## Compensation for MS2-MCP dissociation

The photo-conversion measurements were also corrected for the low, but significant, dissociation of MCP-Dendra2 from the RNA loops of *MS2*. The K*off* for MCP bound to the version of the MS2 stem-loop (C-loop) used in our constructs (U5C) is 0.0017\*min$^{-1}$ (*Lowary and Uhlenbeck, 1987*), which translates into a loss of 9% of the signal after 55 min. To compensate for this dissociation, we applied the following single exponential to the RNA photo-conversion data (already normalised for photobleaching during image acquisition and after removal of the nonspecific component):

$$MI(t) = NI_t / e^{-(p*t)} \tag{7}$$

where $MI(t)$ is the fluorescence intensity normalised for MS2-MCP dissociation as a function of time $(t)$, $NI_t$ is the RNA fluorescence intensity at $t$ time, and $p$ is the 0.0017\*min$^{-1}$ off-rate ($K_{off}$) between MCP and the MS2 C-loop.

## Stimulated emission depletion (STED) super-resolution imaging

Super-resolution imaging was performed on a custom STED microscope built around the IX83 Olympus frame (Olympus, Japan). The microscope design is a variant of a STED system described in detail previously (*Bottanelli et al., 2016*). Imaging was performed with either a 100x UPlanSApo 1.4 NA oil immersion objective lens (Olympus, Japan) or a 100x UPlanSApo 1.35 NA silicone oil immersion objective lens (Olympus, Japan) over a region of 15 × 15 µm with square pixels of 14.6 nm (1024 × 1024 pixels).

## Estimation of mRNA particle size, fluorescence intensity and mean distance to nearest neighbour

To measure the size and fluorescence intensity of RNA particles, as well as their mean distance to the nearest neighbour, we created Profiler, a plugin for Fiji that maps intensity maxima in a 32-bit STED image and fits a Lorentzian function to the X-axis and Y-axis line profiles centred on each maximum using the ImageJ Minimizer class. The Lorentzian function was chosen as the best approximation of the STED system Point Spread Function. The plugin Graphic User Interface allows the definition of the profile radius, which was set to 146 nm (10 pixels), the profile width across which

the mean intensity values are taken, which was set to 3 pixels, noise tolerance for maxima detection, which was set to 2. The full width at half maximum (FWHM) of each curve was used as an estimate for the corresponding particle diameter. Noise and touching objects were excluded based on the $r^2$ values of the fitted functions being <0.8 and the ratio between X-axis and Y-axis FWHMs being >2. Regions of high particle density were excluded because the optical sections were larger than the average RNA particle size (~700 nm *versus* 70–120 nm), which means that several particles can be superimposed along the z-axis, leading to overestimation of fluorescence intensities and flawed curve fittings.

The summed fluorescence intensity of each particle was used as read-out of mRNA content. Each particle area was defined as the circle/oval area fitting the X-axis and Y-axis FWHMs, over which the background-subtracted summed fluorescence was calculated. Only datasets from images acquired with equivalent, and thus comparable, parameters of laser power and data acquisition were analysed.

The RNA particle properties were not analysed in nurse cells because of the small number of particles detected.

A mixed-effects linear model was used for the statistical comparison of the mRNA particle sizes and summed fluorescence intensities, with mRNA or genotype as the fixed effect variable and variability between oocytes as the random effect variable. The effect of distance from the oocyte anterior on the particle size and summed fluorescence was evaluated by applying a mixed-effects linear model, with distance from the anterior of the oocyte as the fixed effect variable and the variability beween oocytes as the random effect variable. All statistical tests were performed on the software R (R Project for Statistical Computing, RRID:SCR_001905) (*R Core Team, 2013*).

Profiler also includes a 'clumps' mode to measure the mean distance to the nearest neighbour of intensity maxima within a region of interest, which was used to estimate the proximity of particles in particle-dense areas.

## Two-colour STED imaging

GFP-labelled mRNAs were immuno-stained with ATTO647N-coupled GFP nanobodies (Chromotek, Germany), whereas F-actin or P-bodies were stained with ATTO590 (ATTO590-conjugated Phalloidin, Sigma-Aldrich, MO, USA, or anti-Me31B primary mouse antibody followed by ATTO590-conjugated anti-mouse secondary antibody, ATTO-TEC, Germany). These were then imaged sequentially, using either a 590 nm or a 640 nm laser to excite each fluorophore. Because STED intrinsically leads to bleed-through of shorter wave-length fluorescence into longer wavelength channels, we applied the Spectral Unmixing plugin for Fiji (Joachim Walter; http://rsb.info.nih.gov/ij/plugins/spectral-unmixing.html) to retrieve only the specific signal from each component. The unmix matrices were generated by imaging samples excited with the 590 nm laser but only labelled with ATTO647N or excited with the 647 nm laser but only labelled with ATTO590.

## **Reverse transcriptase and real-time quantitative PCR**

Total RNA was extracted from the ovaries of twenty 48–72 hr old females using the RNeasy kit (Qiagen, Germany). 100 ng of total RNA was then reverse-transcribed using the qPCRBIO cDNA Synthesis Kit (PCR Biosystems, UK), using a combination of poly-dT and random hexanucleotide primers. Real-time PCR was performed on the reverse transcribed samples to independently amplify two regions in the *bcd* mRNA 3'UTR as well as one region in the internal control, *DHFR* RNA. The primer pairs used were:

bcd 3'UTR 1: 5'-GATGTATCTGGGTGGCTGCT-3' & 5'-CCGAAATGTGGGACGATAAC-3'
bcd 3'UTR 3: 5'-CACTAAAGCCCGGGAATATG-3' & 5'-TTTCTTGCTGGCTCGGAATA-3'
DHFR: 5-CTGAGCACCACACTTCAGGA-3' & 5-TGGTAATGTACAGCCGGTGA-3'

Amplifications were performed using the qPCRBIO SyGreen Mix Hi-ROX Kit (PCR Biosystems, UK) and the StepOne Plus Real-Time PCR system (Applied Biosystems, CA, USA).

The relative amounts (fold change) of *bcd* RNA in samples were then quantified by the comparative $C_T$ method (*Schmittgen and Livak, 2008*), using the threshold cycles ($C_T$) calculated by the inbuilt StepOne Real-Time PCR software (Applied Biosystems, CA, USA, StepOne Software, RRID: SCR_014281):

$$\text{Fold change} = 2^{-\Delta\Delta C_T} = 2^{[(C_T \text{ gene of interest} - C_T \text{ internal control})_{\text{sampleA}} - (C_T \text{ gene of interest} - C_T \text{ internal control})_{\text{sampleB}}]} \quad (8)$$

Quantitations represent three biological replicates and two technical replicates, and were performed on the softwares Excel (Microsoft, CA, USA) and R (R Project for Statistical Computing, RRID:SCR_001905) (*R Core Team, 2013*).

## Acknowledgements

We would like to thank Jordan Raff, Uri Abdu and Trudi Schupbach for fly stocks, Anne Ephrussi for the *FRT2A*, *Ge-1*$^{\Delta5}$ fly stock and the *osk* rescue construct, Akira Nakamura for the anti-Me31B monoclonal antibody, Philip Khuc Trong, George Allen and the Cambridge University Statistics Clinic for general assistance with statistical analysis, Natalia Bulgakova for advice on the analysis of FRAP and photo-conversion experiments and the Gurdon Institute Imaging Facility for general assistance with microscopy.

This work was supported by a Wellcome Trust Principal Fellowship to D St J (080007), by a BBSRC/EURORNAQ grant (BB/F010303) and by core support from the Wellcome Trust (092096) and Cancer Research UK (A14492). VT was supported in part by an EU Marie Curie Intra-European Fellowship (236621), KB was supported by a Darwin Trust Scholarship and GS by a Wellcome Trust Strategic Award (095297).

## Additional information

### Funding

| Funder | Grant reference number | Author |
|---|---|---|
| European Commission | Seventh Framework Programme (FP7), Marie Curie Intraeuropean felowship, 236621 | Vítor Trovisco |
| Wellcome Trust | Principle Fellowship (080007) | Vítor Trovisco Daniel St Johnston |
| Darwin Trust Scholarship | | Katsiaryna Belaya |
| Wellcome Trust | Strategic Award, 095297 | George Sirinakis |
| Biotechnology and Biological Sciences Research Council | BBSRC/EURORNAQ grant (BB/F010303) | Daniel St Johnston |
| Wellcome Trust | 092096 | Daniel St Johnston |
| Cancer Research UK | A14492 | Daniel St Johnston |

The funders had no role in study design, data collection and interpretation, or the decision to submit the work for publication.

### Author contributions

VT, Conception and design, Performed most of the experiments and analysed the data, drafted and revised the manuscript with DStJ; KB, Started the project, performed and analysed the initial *bcd* mRNA particle tracking experiments; DN, Performed the analysis of RNA localization and microtubule cytoskeleton organisation in the *shot*$^{2A2}$ mutant, provided the Patronin-YFP and mCherry-Patronin transgenics prior to publication, drafting or revising the article; UI, Created the *bcdMS2* transgene, Commented on the manuscript; GS, Built the STED microscope and assisted with the collection and analysis of the super-resolution data; RB, Assisted with the analysis of the bcd mRNA particle tracking data, assisted with the analysis of the co-localization data using his unpublished Fiji plugin EsCOP, wrote the Fiji plugin Profiler for the analysis of RNA particle properties from the super-resolution images; JJL, Created the MCP-Dendra2 transgenic line; ERG, Designed the MCP-Dendra2 construct, made extensive comments on the manuscript and revisions; DSJ, Conceived and designed the project with VT and KB, analysed and interpreted the data with VT, drafted and revised the manuscript with VT

## Author ORCIDs

Vítor Trovisco, http://orcid.org/0000-0002-8728-0281
Uwe Irion, http://orcid.org/0000-0003-2823-5840
Daniel St Johnston, http://orcid.org/0000-0001-5582-3301

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
