## [Decision Letter]

Thank you for submitting your article "bicoid mRNA localises to the *Drosophila* oocyte anterior by random transport and anchoring" for consideration by *eLife*. Your article has been reviewed by two peer reviewers, and the evaluation has been overseen by Mani Ramaswami as Reviewing Editor and K VijayRaghavan as the Senior Editor. The following individuals involved in review of your submission have agreed to reveal their identity: Ilan Davis (Reviewer #1).

The reviewers have discussed the reviews with one another and the Reviewing Editor has drafted this decision to help you prepare a revised submission.

Summary:

The manuscript by Trovisco et al., characterises the mechanism of bicoid (*bcd*) mRNA localisation in stage 9 oocytes. The authors use the MS2/MCP system to visualize bicoid RNA localization and several fast and high resolutions methods. These methodological advances allow the authors to detect some new aspects of bicoid localization that have previously not been addressed including. the demonstration that during the early stage only microtubules and dynein motors appear to be important for localization, and that bicoid RNA assembles into particles that may be important for its firm tethering at the anterior cortex.

They show that *bcd* mRNA is actively transported by Dynein along MTs (Figure 1). Although expected and previously shown to some extent, this work demonstrated this more definitively to a standard comparable to the previous work by the group on *osk*. They show that there is only a mild directionality bias in the direction of *bcd* mRNA particle travel at the anterior (52.6%:47.4%, compared with 40%:60% for *osk* at the posterior). Dynein is required for the transport, whereas Kinesin 1 (Kin 1) is effectively required to reduce the speed (speed increases when Kin 1 is mutated). In shot mutants (where MT polarity is abolished) *bcd* mRNA localisation is normal but hts mRNA localisation is disrupted (Figure 2). Localised *bcd* mRNA is anchored at the anterior (Figure 3) and this anterior anchoring is not MT dependent (Figure 4), nor is *bcd* directly anchored on cortical actin (Figure 5). *bcd* mRNA is partially localised to P bodies. Finally, they use STED super resolution imaging to measure the full width half max size of *bcd* mRNA particles, discovering an unexpectedly consistent particle size.

While many of the 'negative results', i.e only partial requirement for dynein and microtubules for localisation are supported by genetic data, other results are more correlative. The most interesting finding is that upon arrival to the anterior part of the oocyte via an active, dynein and microtubule dependent mechanisms, bicoid mRNA is stably anchored to the very anterior cortex of the oocyte. This stable anchoring and not continuous active transport via microtubules by motors is apparently keeping bicoid associated at the anterior during the early stages. Importantly, the authors find that bicoid coalesce into mRNA aggregates. This finding is potentially relevant to previous results by a number of groups (Nüsslein-Volhard, Brunel, MacDonald) that showed that bicoid RNA dimers are necessary for bicoid localization. In particular, the Macdonald lab showed a specific requirement for Loop 3 which contains the dimerization motif. The subject of possible 'protein-supported- RNA self-organization' as an anchoring mechanism is topical as similar mechanisms were recently proposed for posterior localised RNAs. Unfortunately, the study does not address this later question with the stringency they used to address microtubule-mediated transport.

Essential revisions:

Two lines to additional experimentation will greatly improve the manuscript. Of these, one is essential. For reasons elaborated on below, the emphasis on the particles, which is new, is incomplete and could be misleading without further experiments using FISH experiments to quantitative analysis of number of RNA molecules/particle.

1) The authors conclude that bicoid RNA could be retained at the anterior by aggregating into larger, less diffusive bicoid particles. They quantify the particle size by measuring the FWHM of the fluorescent spots, which allowed them to assess whether the particles contain more or less bicoid mRNAs. However, the only way the number of mRNAs per spot can be adequately determined is through a summed fluorescent intensity measurement. One mRNA can become more or less compacted, it will change its size but the mRNA fluorescent count will remain the same. Doubling the amount of labeled bicoid will not address this problem. Further, extra labeled bicoid should not affect the kinetics of photoconversion since simply more bicoid will be photoconverted but the decay rate constant will remain the same. Otherwise, one could expect that a 100% photoconversion and 50% photoconversion would give different results and that is not true. So these test are not themselves evidence that bicoid does not aggregate.

2) The issue seems further confusing when the authors say: "bicoid mRNA forms larger aggregates at stage 14 of oogenesis, but super-resolution imaging revealed that these are still composed of small, discrete RNA particles. Although the fluorescence curve fitting cannot be applied on clusters due to the skewing affects from neighboring particles, the estimated size of isolated bicoid mRNA particles was comparable to those in stage 9 oocytes." Both of these cannot be correct. If particles are discrete, as the authors observe, then they can be fitted to a curve to measure their size/intensity. Otherwise, they need to revise their statement or measure particle intensities. Also, please show data for: bicoid mRNA forms larger aggregates at stage 14 of oogenesis, but super-resolution imaging revealed that these are still composed of small, discrete RNA particles.

3) Additionally, in Figure 7 the two graphs plot the analysis of the same particles. However, the graphs show different number of particles. Please explain. Graph in 7H shows that the number of mRNAs per particles is not the same depending on the distance from the anterior and that it can change between 3 to up to 5 fold the closer the particle is to the anterior. This result implies bicoid mRNA aggregation varies, contrary to what the authors are proposing. Again, It would be important to establish the detection baseline of a single bicoid mRNA. Could it be that the authors are observing only the brightest/biggest bicoid particles at the anterior, those composed of multiple, coalesced bicoid mRNAs and mistaken them for single bicoid mRNAs? Is there any possibility that the tagging with MS2/MCP affects imaging of particles? Single molecule FISH experiments could resolve this question. Further, there is WT, unlabeled bicoid present in these experiments which means that there could be a substantial "dark" fraction of bicoid mRNA aggregated with labeled bicoid that would appear as single bicoid mRNA but would in fact contain multiple bicoid transcripts, those labeled and those unlabeled. What is the ratio between WT and labeled bicoid mRNA? Are all labeled bicoid mRNA molecules detected?

4) The authors also observe later stages, and the apparent changes in behavior of bicoid particles is astonishing. Recent data from the Davis group suggest that the late phases of bicoid RNA localisation is critical for localization of bicoid. It would be helpful to provide a schematic model for bicoid localization through oogenesis. It would help the reader manoeuvre through the different mechanisms of bicoid localisation through oogenesis.

[Editors' note: further revisions were requested prior to acceptance, as described below.]

Thank you for resubmitting your work entitled "bicoid mRNA localises to the *Drosophila* oocyte anterior by random Dynein-mediated transport and anchoring" for further consideration at *eLife*. Your revised article has been favorably evaluated by K VijayRaghavan (Senior editor), Mani Ramaswami (Reviewing editor), and two reviewers.

The manuscript has been improved. In particular it is very useful to see the dosage experiments revealing the consistency of RNA granule size independent of RNA quantity. However, there are some remaining issues in the presentation that need to be addressed before acceptance, as outlined below:

1) The manuscripts states that "i) Few if any *bcd* mRNA particles can be detected in nurse cells, suggesting that single RNA molecules that are exported from the nurse cell nuclei are too faint to detect above background." The sentence should be more specific because when smFISH works as well as it can, one should be able to count individual transcripts, if they are in a sufficiently low density. So the text should better explain the possible reasons for why the authors cannot detect bicoid particles in nurse cells – perhaps in the Materials and methods?. Presumably bicoid is expressed at that time? Presumably, their particular probes are not working as well as they could, or their background is too high, so single molecule detection is not possible in their case.

2) The authors state that "We maintain that estimating RNA particle properties in dense or clustered areas is fundamentally flawed – especially fluorescence intensities – due to the thickness of the optical sections and low resolution in the Z dimension (700 nm), which means that several particles can be superimposed in the same optical section." This statement as it is too absolute in nature. It is certainly possible to measure the intensity of RNA particles and estimate the number of mRNA molecules in many cases, and this has been done by various other researchers. Again, it is more likely that the quality of the probe or the presence of high background is the reason why they cannot measure the intensity of bicoid particles reliably in this case. Again, the text could simply state in the methods section that in their particular case they could not achieve this. But, it does not seem to be correct to generalise from their particular case in such an absolute manner.

3) In the last sentence of the paragraph, the authors state that: "These virus-like particles are smaller in *exu* mutants and may package the RNA for transport and anchoring".

This statement seems to be based on data presented in Table 5 and the statistical significance of this difference is p=0.046. This significance seems too borderline for the authors to draw such a strong conclusion and to put such a statement in the Abstract. The idea that this is like a virus particle is rather far fetched to be included in the Abstract. 100 nm is a huge size in molecular terms and while it is ok to suggest that the RNA particle is virus-like, there is not enough evidence to justify including this possibility as a main point in the Abstract.

---

## [Author Response]

Summary:

The manuscript by Trovisco et al., characterises the mechanism of bicoid (bcd) mRNA localisation in stage 9 oocytes. The authors use the MS2/MCP system to visualize bicoid RNA localization and several fast and high resolutions methods. These methodological advances allow the authors to detect some new aspects of bicoid localization that have previously not been addressed including. the demonstration that during the early stage only microtubules and dynein motors appear to be important for localization, and that bicoid RNA assembles into particles that may be important for its firm tethering at the anterior cortex.

They show that bcd mRNA is actively transported by Dynein along MTs (Figure 1). Although expected and previously shown to some extent, this work demonstrated this more definitively to a standard comparable to the previous work by the group on osk. They show that there is only a mild directionality bias in the direction of bcd mRNA particle travel at the anterior (52.6%:47.4%, compared with 40%:60% for osk at the posterior). Dynein is required for the transport, whereas Kinesin 1 (Kin 1) is effectively required to reduce the speed (speed increases when Kin 1 is mutated). In shot mutants (where MT polarity is abolished) bcd mRNA localisation is normal but hts mRNA localisation is disrupted (Figure 2). Localised bcd mRNA is anchored at the anterior (Figure 3) and this anterior anchoring is not MT dependent (Figure 4), nor is bcd directly anchored on cortical actin (Figure 5). bcd mRNA is partially localised to P bodies. Finally, they use STED super resolution imaging to measure the full width half max size of bcd mRNA particles, discovering an unexpectedly consistent particle size.

While many of the 'negative results', i.e only partial requirement for dynein and microtubules for localisation are supported by genetic data, other results are more correlative. The most interesting finding is that upon arrival to the anterior part of the oocyte via an active, dynein and microtubule dependent mechanisms, bicoid mRNA is stably anchored to the very anterior cortex of the oocyte. This stable anchoring and not continuous active transport via microtubules by motors is apparently keeping bicoid associated at the anterior during the early stages. Importantly, the authors find that bicoid coalesce into mRNA aggregates. This finding is potentially relevant to previous results by a number of groups (Nüsslein-Volhard, Brunel, MacDonald) that showed that bicoid RNA dimers are necessary for bicoid localization. In particular, the Macdonald lab showed a specific requirement for Loop 3 which contains the dimerization motif. The subject of possible 'protein-supported- RNA self-organization' as an anchoring mechanism is topical as similar mechanisms were recently proposed for posterior localised RNAs. Unfortunately, the study does not address this later question with the stringency they used to address microtubule-mediated transport.

We appreciate all the reviewers’ comments and corrections and believe that the new data that we have added to address them have significantly improved our manuscript. Here we discuss our responses to the main issues raised by the referees, before addressing all of the more specific points in turn:

i) Few if any *bcd* mRNA particles can be detected in nurse cells, suggesting that single RNA molecules that are exported from the nurse cell nuclei are too faint to detect above background.

ii) It is well known that *bcd* mRNA can dimerize or even multimerise (Ferrandon et al., 1997; Wagner, 2003; Wagner et al., 2001)

iii) A recently published article shows that Exuperantia (Exu) dimerizes, and that this is required for high affinity RNA binding and *bcd* mRNA localization (Lazzaretti et al., 2016). This work on Exu provides support for our suggestion that Exu is required for the proper formation of *bcd* mRNA particles by providing a scaffold for multiple *bcd* mRNA transcripts. The manuscript has been revised to include the findings by Lazzaretti and colleagues.

We agree with the issues raised by the reviewers on our quantitative analyses of the RNA content of the particles labelled with MS2/MCP system, namely that we are not measuring total RNA because the endogenous RNA is not labelled and that the mean intensity of the curve fitting is not the best measure of relative molecule content. We have therefore performed single molecule FISH to detect all *bcd* RNA and imaged the particles also using STED. This allowed us to measure the summed fluorescence intensities as a better measure of the relative RNA content of the particles. The smFISH data gave very similar results on particle sizes to those using MS2/MCP, revealing that *bcd* RNA particles remain the same size both before and after their localisation to the anterior, and at stage 9 and stage 14, when they cluster into large aggregates. They even remain the same size when the levels of *bcd* RNA are increased threefold by expressing two extra copies of *bcd*-MS2.

Importantly, the smFISH analysis of the summed fluorescent intensities reveals that the RNA content of the particles increases when *bcd* RNA is present at higher levels, both when extra copies of *bcd* are added, and at stage 14, after nurse cell dumping has delivered much more *bcd* RNA to the oocyte (Weil et al., 2006). Thus, the particles have a fixed size regardless of how much RNA they contain, suggesting that they provide a fixed scaffold that can recruit a variable number of *bcd* RNA molecules, depending on the RNA concentration.

We disagree with the reviewers’ comment (2) that we should be able to measure the intensities and sizes of the *bcd* RNA particles in dense clusters at stage 14 because we can see them. We maintain that estimating RNA particle properties in dense or clustered areas is fundamentally flawed – especially fluorescence intensities – due to the thickness of the optical sections and low resolution in the Z dimension (700 nm), which means that several particles can be superimposed in the same optical section. To illustrate this point, we have attached to this response a figure with the XZ and XZ fluorescence profiles of 100 nm beads imaged with confocal and STED modes. This shows that the STED mode improves the XY-resolution to faithfully represent the 100 nm bead size, but not the Z-profile, which is seven times bigger. Since the average spacing between particles (measured in the XY dimension) in these clusters is 204 nm, we are likely to be imaging several particles in the Z axis. This makes the measurement of the RNA content of the particles using summed fluorescence intensities misleading. This problem also impairs the measurement of particle size because the background signal between peaks is higher. Nevertheless, we used the smFISH data to measure the particles sizes in the clusters at stage 14, as requested by the referees, and found no significant difference between the sizes of the clustered particles and the isolated particles (or particles at stage 9).

All the new analyses and experiments of STED imaging of RNA particles led to significant changes to Table 5 and Figure 7, which we have now split into Figure 7, which includes the microscopy images and all the analyses of the particle sizes, and Figure 8, which shows the analyses on the summed fluorescence intensities of the particles.

Essential revisions:

Two lines to additional experimentation will greatly improve the manuscript. Of these, one is essential. For reasons elaborated on below, the emphasis on the particles, which is new, is incomplete and could be misleading without further experiments using FISH experiments to quantitative analysis of number of RNA molecules/particle.

We thank the reviewers for encouraging us to do FISH experiments, which we think have significantly enhanced our analysis.

1) The authors conclude that bicoid RNA could be retained at the anterior by aggregating into larger, less diffusive bicoid particles.

To clarify, we considered the hypothesis that the retention of *bcd* RNA at the oocyte anterior could be due to the fusion of particles by dimerisation/oligomerisation. We rejected this hypothesis because we find that the particles remain the same size with the same RNA content when they reach the anterior. Some other mechanism must therefore retain the particles at the anterior.

They quantify the particle size by measuring the FWHM of the fluorescent spots, which allowed them to assess whether the particles contain more or less bicoid mRNAs. However, the only way the number of mRNAs per spot can be adequately determined is through a summed fluorescent intensity measurement. One mRNA can become more or less compacted, it will change its size but the mRNA fluorescent count will remain the same.

We agree with the reviewers that the summed fluorescence intensity is a better estimate of the mRNA content of the particles than the mean fluorescence intensity of the X and Y profile curve fits.

Analysis of the summed fluorescence intensities in the original *bcd**GFP datasets revealed no statistical difference between particles in oocytes expressing 1 or 2 copies of *bcdMS2* (in addition to the endogenous copies of non-tagged RNA), and no statistical effect of the distance from the anterior in oocytes expressing 1 *bcdMS2* copy. However, the single molecule FISH experiments allowed us to detect all RNA and therefore to compare the summed fluorescence intensities of particles in oocytes expressing the two endogenous copies of *bcd* with those expressing one or two extra copies of *bcdMS2*. We also quantified the degree of over-expression produced by these transgenes as 1.7 fold and ~3 fold (Figure 7). There was still no statistical difference between the 1x and 2x *bcdMS2* samples, in agreement with *bcd**GFP datasets, but we observed a significant increase in the 1x and 2x *bcdMS2* samples compared to wildtype. We also analysed the summed fluorescence intensities of endogenous *bcd* mRNA particles in stage 14 oocytes and found these contain more than 3x more RNA than those in stage 9 oocytes. Again the size of the particles remained constant.

These combined results lead to conclusion that *bcd* RNA forms particles of defined size, but with variable RNA content, which increases when more RNA is expressed and at later stages of oogenesis, when cytoplasmic dumping delivers large amounts of *bcd* RNA into the oocyte from the nurse cells.

The addition of the summed fluorescence intensity measurements has prompted us to revise Table 5 and Figure 7, which is now divided into two Figures: Figure 7 – images and the analysis of the particle sizes –, and Figure 8 – analysis of the summed fluorescence intensities of the particles. The main text has been also revised to include the new methodology, results and conclusion:

Results section: “To determine whether the particles also contain the same amount of RNA throughout oogenesis and at different *bcd* gene dosages, we used the summed fluorescent intensities of the particles as measure of their RNA content. The average fluorescent intensity of the particles detected by both MS2-labelling and smFISH did not change with distance from the anterior, reinforcing the conclusion that the particles do not fuse upon localisation and anchoring at the anterior (Figure 8). By contrast, the mean fluorescence intensity of the particles increased when *bcd* mRNA levels were raised by expressing one or two copies of the *bcdMS2* transgene, with 30% and 60% more fluorescence, and thus more RNA, per particle respectively (Figure 8, Table 5). Furthermore, stage 14 particles contained more than 3 times as much RNA as those at stage 9 (Figure 8, Table 5). It has previously been shown that most *bcd* RNA enters the oocyte during nurse cell dumping at stage 10b-12 (Weil et al. 2006), suggesting that much of this extra RNA is incorporated into pre-existing RNA particles. Thus, the *bcd* RNA particles have a variable RNA content, despite their constant size.”

Conclusions:” Importantly, over-expression of the mRNA does not alter particle size, but instead leads to more particles, which have higher average RNA content. The same occurs at stage 14 of oogenesis, when the *bcd* mRNA content of the oocyte is much higher following nurse cell dumping. Thus, *bcd* mRNA seems to assemble into a structure of defined size, almost like a virus particle, which can incorporate more or less mRNA molecules depending on availability.”

Conclusions: “By contrast, *bcd* RNA particles stay the same size as the RNA concentration increases, even though they incorporate more RNA, and they do not appear to fuse when tightly clustered in aggregates. The exact nature of the particles will require the identification of more of their components, but their behaviour is compatible with a model in which they consist of a rigid protein framework that contains multiple RNA-binding sites.”

Material and methods section: “Estimation of mRNA particle size, fluorescence intensity & mean distance to nearest neighbour:

To measure the size and fluorescence intensity of RNA particles, as well as their mean distance to the nearest neighbour, we created Profiler,…”.

Material and methods section: “The summed fluorescence intensity of each particle was used as read-out of mRNA content. For that, each particle area was defined as the circle/oval area fitting the X-axis and Y-axis FWHMs, over which the background-subtracted summed fluorescence was calculated. Only datasets from images acquired with equivalent, and thus comparable, parameters of laser power and data acquisition were analysed.

A mixed-effects linear model was used for the statistical comparison of the mRNA particle sizes and summed fluorescence intensities, with mRNA or genotype as the fixed effect variable and oocyte as the random effect variable. The effect of distance from the oocyte anterior on the particle size and summed fluorescence was evaluated by applying a mixed-effects linear model, with distance from the anterior of the oocyte as the fixed effect and oocyte as the random effect”

Doubling the amount of labeled bicoid will not address this problem. Further, extra labeled bicoid should not affect the kinetics of photoconversion since simply more bicoid will be photoconverted but the decay rate constant will remain the same. Otherwise, one could expect that a 100% photoconversion and 50% photoconversion would give different results and that is not true. So these test are not themselves evidence that bicoid does not aggregate.

There seems to be some confusion about the rationale behind the experiments with extra copies of *bcd*. We performed these experiments to address whether the mRNA particle size depends on RNA concentration. If so, one would expect that the particles would become bigger and/or brighter as more copies of *bcd* are added. We hope that the more thorough analysis that we have now added makes clear that the particles get brighter, but not bigger, as the RNA concentration is increased.

The fluorescence decay upon photo-conversion should only remain the same if the ‘diffusion’ properties of the particles also remain the same. If over-expressing *bcd* mRNA promoted the formation of larger particles, one would expect them to diffuse more slowly, which should also slow down the decay of red fluorescence after photo-conversion. Our data (Figure 7—figure supplement 1) indicate that this is not the case, which is consistent with our observation that the particles do not change size as the RNA dosage is increased.

2) The issue seems further confusing when the authors say: "bicoid mRNA forms larger aggregates at stage 14 of oogenesis, but super-resolution imaging revealed that these are still composed of small, discrete RNA particles. Although the fluorescence curve fitting cannot be applied on clusters due to the skewing affects from neighboring particles, the estimated size of isolated bicoid mRNA particles was comparable to those in stage 9 oocytes." Both of these cannot be correct. If particles are discrete, as the authors observe, then they can be fitted to a curve to measure their size/intensity. Otherwise, they need to revise their statement or measure particle intensities.

The reviewers are correct that if particles are discrete, as evident in Figure 7, one should be able to fit curves to their intensity profiles using the same algorithm. However, we believe such an analysis is fundamentally flawed, because the mean distance between nearest neighbours in clustered/dense areas of stage 9 and stage 14 oocytes, (224 nm and 202 nm respectively) is much less than the thickness of the optical sections along the Z axis (~700 nm). In dense areas, each optical section can therefore include up to 4 particles lying above and below each other. This means that the summed intensities are not measurements from single particles. It also leads to an irregular background that skews the apparent particle sizes, as discussed above in our general response. We have revised the text to discuss this issue and have added the data and methodology for the determination of the mean distance to nearest neighbour:

Results section: “As the particles in clusters are about 200 nm apart in the XY dimension (224 nm and 202 nm mean distance to nearest neighbour in stage 9 and 14 oocytes, respectively), each ~700 nm optical Z-section is likely to include more than one particle. This causes an irregular background that makes fluorescence intensity curve fitting unreliable. Nevertheless, the estimated size of isolated *bcd* mRNA particles at stage 14 was comparable to those in stage 9 oocytes (Table 5, Figure 7).”

Material and methods section: “Profiler also includes a ‘clumps’ mode to measure the mean distance to the nearest neighbour of intensity maxima within a region of interest, which was used to estimate the proximity of particles in particle-dense areas.”

Also, please show data for: bicoid mRNA forms larger aggregates at stage 14 of oogenesis, but super-resolution imaging revealed that these are still composed of small, discrete RNA particles.

We thank the reviewers for bringing this to our attention and apologise for not citing Figure 7 in the text. We have changed the text as follows: “*bcd* mRNA forms larger aggregates at stage 14 of oogenesis, but super-resolution imaging revealed that these are still composed of small, discrete RNA particles (Figure 7).”

Figure 7 has now been significantly changed, Panel C now includes the corresponding confocal and super-resolution STED images of stage 14 *bcd**GFP particle clusters. We hope that this will make it clear that the large aggregates seen in the confocal image are composed of much smaller individual particles when examined with super-resolution. Since we have now performed STED imaging of smFISH-labelled endogenous *bcd* RNA, we also included the corresponding images for stages 9 and 14 as panels E and F. We also now explain in the legend to Figure 7 that the insets in panels A and C are the corresponding confocal (left) and super-resolved (right) images.

3) Additionally, in Figure 7 the two graphs plot the analysis of the same particles. However, the graphs show different number of particles. Please explain.

The reason for using different datasets was that Figure 7 data were from images obtained using variable parameters of laser intensity and data acquisition, which renders them unsuitable for fluorescence intensity comparison, whereas Figure 7 data were from images obtained under identical parameters.

Following the reviewers’ suggestion, we have performed STED imaging on endogenous bicoid mRNA using single molecule FISH and have analysed both particle size and fluorescence intensity in the same dataset. We therefore decided to split Figure 7 into Figure 7 and 8. Figure 7 includes the particle size analyses. The previous Figure 7 is now Figure 7 and we have included a similar size scatterplot for the new smFISH data on endogenous *bcd* RNA particles in stage 9 oocytes – Figure 7.

The equivalent to the previous Figure 7 is now Figure 8 and includes the summed fluorescence intensity data on an enlarged dataset of *bcd**GFP particles in stage 9 oocytes expressing 1x *bcdMS2*. We have also included a scatterplot showing the summed fluorescence intensities of endogenous *bcd* RNA particles at stage 9 from the new smFISH data as Figure 8.

Graph in 7H shows that the number of mRNAs per particles is not the same depending on the distance from the anterior and that it can change between 3 to up to 5 fold the closer the particle is to the anterior. This result implies bicoid mRNA aggregation varies, contrary to what the authors are proposing.

The decrease in the mean particle summed fluorescence intensity with the distance from the anterior was not statistically significant (p=0.19, bottom-right box). The new summed fluorescence analyses of both *bcd**GFP and endogenous *bcd* RNA (i.e smFISH) also yielded a non-statistically summed fluorescence with the distance from the anterior (p=0.11 and p=0.13, Table 5; new Figure 8). Thus, both the old and new data support the idea that the amount of *bcd* mRNA per particle does not change upon localization.

The non-significant increase in summed fluorescence intensity closer to the anterior is probably due to the higher particle density in this region and the corresponding increase in the probability that a fluorescent peak is composed of two or more particles above each other along the Z-axis (see above). The second source of variability in the apparent size and intensity of individual particles is the distance between a particle’s centre and the optical focus, which will change the fluorescence distribution depending on the PSF. This is the case for all quantitative analyses of this type, which is why one needs to analyse hundreds of particles to obtain statistically robust measurements. Thus, the variability does not reflect variation in the actual size of particles, but in the degree to which they are in focus and are super-imposed.

Curiously, smFISH for endogenous bicoid mRNA (*yw* genotype) yielded a minor (but statistically significant decrease in particle sizes with proximity to the anterior, which is the opposite of what one would expect if they fuse upon localisation. We do not know the reason for this change or why we only observe it with FISH, but the effect is so small (1.7 nm/µm) that it cannot reflect particle splitting, which would induce a change of 20% in size assuming that the total volume is conserved. This may instead reflect a small increase in the compaction of the RNA at the anterior or may be due to the effect of the higher density on the curve fitting method (new Figure 7). In either case, it cannot be accounted for a major remodelling of the particles, such as merging.

Again, It would be important to establish the detection baseline of a single bicoid mRNA. Could it be that the authors are observing only the brightest/biggest bicoid particles at the anterior, those composed of multiple, coalesced bicoid mRNAs and mistaken them for single bicoid mRNAs? Is there any possibility that the tagging with MS2/MCP affects imaging of particles? Single molecule FISH experiments could resolve this question. Further, there is WT, unlabeled bicoid present in these experiments which means that there could be a substantial "dark" fraction of bicoid mRNA aggregated with labeled bicoid that would appear as single bicoid mRNA but would in fact contain multiple bicoid transcripts, those labeled and those unlabeled.

We agree with this comment, but never claimed that the particles represent single molecules of *bcd* mRNA. It is very likely that each particle is formed by a number of mRNA molecules, because bicoid mRNA can dimerise and it has recently been published that Exuperantia homo-dimerises via its inactive exonuclease domain and that this dimerisation is important for high affinity RNA binding and *bcd* mRNA localization (Lazzaretti et al., 2016). The findings of this study are now discussed in the revised text:

Introduction: “Only pre-treatment with nurse cell cytoplasm renders in vitro transcribed RNA competent to localise specifically to the oocyte anterior, and this conditioning requires the pseudonuclease Exuperantia (Exu) (Cha et al. 2001). The role of Exu in the localisation of *bcd* mRNA requires its homo-dimerisation and RNA binding (Lazzaretti et al. 2016).”

Results section: “The localisation of *bcd* mRNA at all stages depends on Exu protein (Berleth et al. 1988) and we therefore also examined the behaviour of *bcd* mRNA particles in the mutant, *exu*^1^, which reduces the affinity of Exu to RNA (Lazzaretti et al. 2016).”

Results section: “Thus, Exu is required for both the formation of *bcd* mRNA particles and their efficient transport on microtubules, consistent with the dimerisation of Exu (possibly leading to the dimerisation of *bcd* mRNA) and the results obtained from *bcd* mRNA injections (Cha et al. 2001; Lazzaretti et al. 2016). “

Results section:”Indeed, the only condition that altered the size of the *bcd* mRNA particles was the *exu*^1^ mutant, which strongly reduces the affinity of Exu for RNA (Lazzaretti et al. 2016). In this case, the few particles that were detected were slightly, but significantly, smaller (98 nm versus 112 nm) (Figure 7, Table 5). Thus, Exu may provide part of the scaffolding for the assembly of *bcd* mRNA particles.”

Discussion section: “This suggests that Exu, which forms homodimers that probably bind two *bcd* mRNA molecules (Lazzaretti et al., 2016), plays a role in scaffolding the assembly of the particles (Figure 9).”

Obviously, when using the MS2/MCP-labelling system the endogenous molecules are not labelled and we therefore cannot say whether all particles are labelled and detected. smFISH overcomes this limitation. As already mentioned, we performed STED imaging of *bcd* mRNA labelled by smFISH and obtained very similar results to those from the MS2/MCP system and the text has been amended to include the new smFISH data, conclusions and methodology:

Results section: “To confirm these findings, we also performed STED imaging on endogenous *bcd* RNA labelled by single molecule FISH (smFISH) with probes spanning the 3’UTR. This technique also revealed that *bcd* RNA forms particles that remain approximately the same size throughout oogenesis, although they appear slightly larger than those seen with MS2-GFP labelling, presumably because smFISH labels the entire *bcd* 3’UTR, not just the MS2 sites in the RNA (Figure 7, Table 5).”

Results section “The average size of *bcd* mRNA particles measured by both MS2-GFP-labelling and smFISH did not change substantially with distance from the anterior (Figure 7, Table 5), arguing against their coalescence upon localisation.”

Material and methods section:

“Single molecule FISH (smFISH): Ovaries from 48–72 hr old females were dissected in PBS and fixed in 4% formaldehyde in PBS for 20 min. For stage 14 oocytes, ovaries from 72-96h old virgin females were dissected into Modified Robb’s medium, fixed in Cacodylate Fixative for 20 min (McKim et al. 2009) and washed in PBS. Samples were dehydrated in ethanol at 4ºC for 18h, incubated in wash buffer (2X saline-sodium citrate (SSC), 10% formamide) for 30 min, and then hybridized with 500 nM ATTO647N-conjugated antisense Stellaris probes for *bcd* RNA (Biosearch Technology) in hybridization buffer (10 dextran sulfate, 2X SSC, 10% formamide) at 37ºC for 4 hr. After 2 hr in wash buffer, samples were mounted in Vectashield mounting medium (Vectorlabs).

Antisense RNA probes spanning the 3’UTR of *bcd* RNA:

5’-GAAACTCTCTAACACGCCTC-3’, 5’-ACAGTGGTTAACCTAAAGCT-3’, 5’-TGGTATTTGTACAATCAGGA-3’, 5’-CTTTCTACGCGTAGATATCT-3’, 5’-ACGGATCTTAGGACTAGACC-3’, 5’-AAACTTCCCTGGGAACCATT-3’,

5’-CTGCTGACTAGGCTAGTACA-3’, 5’-GATATGCACTGGAATCCGTG-3’,

5’-GAGTTAACTGGAGTATCACT-3’, 5’-AGCGTATTGCAGGGAAAGTA-3’,

5’-CACCCAGATACATCTAAGGC-3’, 5’-CATATTCCCGGGCTTTAGTG-3’, 5’-TGGCCTCAAATGTAACTGGT-3’, 5’-ACTTTCCATGGAATACGCTT-3’, 5’-ATTTCCGAAATGTGGGACGA-3’, 5’-AGAAGATTTTCTTGCTGGCT-3’, 5’-GTACAGTTTTTAGCTATGTC-3’, 5’-ATGAGATTACGCCCAAGAGA-3’, 5’-ATGTTCGATCTTTAAGGGTA-3’, 5’-ACACTTTGGCATAGCATAGA-3’, 5’-GCGCAAATGTTTGATTATGT-3’, TTGCTGACTATTCTTGGTCA-3’, 5’-ACAAATGGTCTGCATTGATT-3’, 5’-TGATAGTTATTCCGTTTGGC-3’, 5’-ATGCTCTTCTTAGTGATGTA-3’, 5’-ACTTGAGGCCTAACAGATTG-3’, 5’-ACAACATCAAAGGTGCAGCA-3’ & 5’-ATTTACCCGAGTAGAGTAGT-3’.”

What is the ratio between WT and labeled bicoid mRNA? Are all labeled bicoid mRNA molecules detected ?

We measured the ratio of endogenous *bcd* mRNA to transgenic *bcdMS2* RNA by quantitative RT-PCR (qPCR) and have added these data as Figure 7. qPCR indicated that 1 copy of the *bcdMS2* transgene produces nearly the same amount of RNA as the two endogenous copies (0.75), meaning that almost half of the molecules will be labelled. Two copies of the *bcdMS2* transgene produce slightly more than twice the amount of the endogenous genes, labelling 2/3 of the total RNA. This, combined with the fact that the super-resolution analyses of particles labelled by smFISH yielded mostly similar results to the MS2 labelling, shows that MS2/MCP was a robust system for the overall analysis of *bcd* mRNA localisation.

We have added a description of the RT and qPCR methodology to the text:

“Reverse Transcriptase & real-time quantitative PCR

Total RNA was extracted from the ovaries of twenty 48-72h old females using the RNeasy kit (Qiagen). 100 ng of total RNA was then reverse-transcribed using the qPCRBIO cDNA Synthesis Kit (PCR Biosystems), using a combination of poly-dT and random hexanucleotide primers. Real-time PCR was performed on the reverse transcribed samples to independently amplify two regions corresponding to *bcd* mRNA 3’UTR, as well as one region corresponding to the internal control DHFR gene RNA. The primer pairs used were:

*bcd* 3’UTR 1: 5’-GATGTATCTGGGTGGCTGCT-3’ & 5’-CCGAAATGTGGGACGATAAC-3’

*bcd* 3’UTR 3: 5’-CACTAAAGCCCGGGAATATG-3’ & 5’-TTTCTTGCTGGCTCGGAATA-3’

DHFR: 5-CTGAGCACCACACTTCAGGA-3’ & 5-TGGTAATGTACAGCCGGTGA-3’

Amplifications were performed using the qPCRBIO SyGreen Mix Hi-ROX Kit (PCR Biosystems) and the StepOne Plus Real-Time PCR system (Thermo Fisher Scientific).

The relative amounts of *bcd* RNA in samples were then quantified by the comparative CT method (Schmittgen and Livak, 2008), using the threshold cycles (CT) calculated by the inbuilt Applied Biosystems StepOne Real-Time PCR Software v2.3:

Eq.8 Fold change = 2-∆∆CT = 2-[(CT gene of interest – CT internal control)sampleA – (CT gene of interest – CT internal control)sampleB]

Quantitations represent three biological replicates and two technical replicates, and were performed using Excel and R.”

4) The authors also observe later stages, and the apparent changes in behavior of bicoid particles is astonishing. Recent data from the Davis group suggest that the late phases of bicoid RNA localisation is critical for localization of bicoid. It would be helpful to provide a schematic model for bicoid localization through oogenesis. It would help the reader manoeuvre through the different mechanisms of bicoid localisation through oogenesis.

Following this suggestion, we have created Figure 9, with the requested model.

[Editors' note: further revisions were requested prior to acceptance, as described below.]

[…]

1) The manuscripts states that "i) Few if any bcd mRNA particles can be detected in nurse cells, suggesting that single RNA molecules that are exported from the nurse cell nuclei are too faint to detect above background." The sentence should be more specific because when smFISH works as well as it can, one should be able to count individual transcripts, if they are in a sufficiently low density. So the text should better explain the possible reasons for why the authors cannot detect bicoid particles in nurse cells – perhaps in the Materials and methods?. Presumably bicoid is expressed at that time? Presumably, their particular probes are not working as well as they could, or their background is too high, so single molecule detection is not possible in their case.

We analysed 36 images of nurse cells, from which only 51 fluorescence maxima (i.e. putative particles) could be automatically detected, only one of which produced good curve fits. This shows that, as expected, *bcd* mRNA is not abundant in the nurse cells, possibly due to its quick delivery into the oocyte. This also suggests that *bcd* mRNA particles are “too faint to detect above background”.

We included the following statement in the Materials and methods (‘Stimulated Emission Depletion (STED) Super-resolution Imaging’ section, ‘Estimation of mRNA particle size, fluorescence intensity & mean distance to nearest neighbour’ sub-section): “The RNA particle properties were not analysed in nurse cells because of the small number of particles detected.”

2) The authors state that "We maintain that estimating RNA particle properties in dense or clustered areas is fundamentally flawed – especially fluorescence intensities – due to the thickness of the optical sections and low resolution in the Z dimension (700 nm), which means that several particles can be superimposed in the same optical section." This statement as it is too absolute in nature. It is certainly possible to measure the intensity of RNA particles and estimate the number of mRNA molecules in many cases, and this has been done by various other researchers. Again, it is more likely that the quality of the probe or the presence of high background is the reason why they cannot measure the intensity of bicoid particles reliably in this case. Again, the text could simply state in the methods section that in their particular case they could not achieve this. But, it does not seem to be correct to generalise from their particular case in such an absolute manner.

It was never our intention to state that it is absolutely impossible to accurately estimate RNA particle properties in clusters and dense areas but rather that our system does not allow it. The thickness of our optical sections is much bigger than the size of particles (~700 nm vs 70-120 nm), which means that several particles can be superimposed, rendering any measurements in dense areas unreliable. The problem arises because the background is high and irregular. Although we tried to address this issue in the previous version of the manuscript, we attempted to further clarify it in both the Results and Materials and methods sections:

Results section: “This causes a high and irregular background, leading to overestimation of that makes fluorescence intensities and unreliable curve fittings.”

Material and Methods section: We replaced “In addition, to avoid flawed curve fittings due to the contribution of fluorescence from neighbouring in or out of focus particles, regions of high particle density were excluded.” with “Also, regions of high particle density were excluded because the optical sections were larger than the average RNA particle size (~700 nm versus 70-120 nm), which means that several particles can be superimposed along the z-axis, leading to overestimation of fluorescence intensities and flawed curve fittings.”

3) In the last sentence of the paragraph, the authors state that: "These virus-like particles are smaller in exu mutants and may package the RNA for transport and anchoring".

This statement seems to be based on data presented in Table 5 and the statistical significance of this difference is p=0.046. This significance seems too borderline for the authors to draw such a strong conclusion and to put such a statement in the Abstract. The idea that this is like a virus particle is rather far fetched to be included in the Abstract. 100 nm is a huge size in molecular terms and while it is ok to suggest that the RNA particle is virus-like, there is not enough evidence to justify including this possibility as a main point in the Abstract.

We understand the concerns regarding the data from *exu* mutants, which is of borderline significance and have changed the last sentence of the Abstract to “These particles appear to be well-defined structures that package the RNA for transport and anchoring.”